# The replication initiation determinant protein (RepID) modulates replication by recruiting CUL4 to chromatin

Sang-Min Jang[1], Ya Zhang[1], Koichi Utani[1], Haiqing Fu[1], Christophe E. Redon[1], Anna B. Marks[1], Owen K. Smith[1], Catherine J. Redmond[1], Adrian M. Baris[1], Danielle A. Tulchinsky[1] & Mirit I. Aladjem[1]

Cell cycle progression in mammals is modulated by two ubiquitin ligase complexes, CRL4 and SCF, which facilitate degradation of chromatin substrates involved in the regulation of DNA replication. One member of the CRL4 complex, the WD-40 containing protein RepID (DCAF14/PHIP), selectively binds and activates a group of replication origins. Here we show that RepID recruits the CRL4 complex to chromatin prior to DNA synthesis, thus playing a crucial architectural role in the proper licensing of chromosomes for replication. In the absence of RepID, cells rely on the alternative ubiquitin ligase, SKP2-containing SCF, to progress through the cell cycle. RepID depletion markedly increases cellular sensitivity to SKP2 inhibitors, which triggered massive genome re-replication. Both RepID and SKP2 interact with distinct, non-overlapping groups of replication origins, suggesting that selective interactions of replication origins with specific CRL components execute the DNA replication program and maintain genomic stability by preventing re-initiation of DNA replication.

[1] Developmental Therapeutics Branch, Center for Cancer Research, NCI, NIH, Bethesda, MD 20892-4255, USA. Correspondence and requests for materials should be addressed to M.I.A. (email: aladjemm@mail.nih.gov)

Eukaryotic cells create an exact and complete copy of their entire cellular genome precisely once each cell cycle, ensuring that all genetic and epigenetic information is accurately transferred to both daughter cells. In most somatic metazoan cells, DNA replication begins at multiple initiation sites, termed replication origins, on each chromosome[1–3]. In healthy individuals, replication origins are activated in a precise order and their activities are strictly constrained by a series of cell cycle checkpoints that are often relaxed in cancer. Strict regulation of the frequency of replication initiation events is mediated by sequential chromatin binding of a series of proteins that form and activate pre-replication complexes (pre-RCs)[2,4,5]. Pre-RC assembly, known as replication origin "licensing", occurs shortly after the mitotic phase is completed. Prior to the onset of DNA replication, pre-RCs recruit additional proteins and are converted to larger pre-initiation complexes that include substrates for Cdc7/Dbf4-dependent kinase (DDK) and cyclin dependent kinases (CDKs). DDK-mediated and CDK-mediated phosphorylation events activate the MCM2-7 helicase and recruit polymerases and accessory proteins to start DNA replication. Pre-RCs disassemble from chromatin after replication initiates and reassemble after mitosis. The assembly and disassembly of pre-RCs on chromatin is critical for prevention of re-replication of genomic DNA and for preservation of genomic integrity.

A key regulatory switch in the modulation of DNA replication requires activation of the replicative helicase by the same kinase complexes that prevent further assembly of the inactive helicase on chromatin. The onset of replication is preceded by selective and sequential degradation of licensing factors and their facilitators[6]. As replication progresses, high CDK activity prevents the assembly of new complexes after the initial pre-RCs dissociate from replicated chromatin[2]. Although the rules governing the decision to activate particular pre-RCs on specific origins in each cell cycle remain unclear[1,7–10], the temporal separation between the licensing and replication steps ensures that each replication origin cannot initiate replication more than once during each cell division cycle.

Cullin-RING E3 ubiquitin ligases (CRLs) mediate ubiquitination of proteins required for cell cycle control and DNA replication and play key roles in the regulatory interactions that maintain genomic stability[11,12]. CDT1, a licensing factor in pre-RC, is targeted by CRL4 (DDB1-CUL4-RBX1 Cullin-RING ubiquitin Ligase 4) during the transition between the G1 and S phases of the cell cycle, and by CRL1 (SKP1-CUL1-F-box, or SCF) during S and G2 phase[13–16]. In most cells, SCF exhibits much lower CDT1 ubiquitination activity than CRL4. Other CRL4 and SCF substrates, which are sequentially degraded during the S-phase following CDT1 degradation, include the CDK inhibitor p21[CIP1/WAF1], which prevents progression into or through S phase, and the histone methyltransferase SET8, which catalyzes mono-methylation at histone H4 lysine 20 residue[17–24]. Dysfunction of CRL4 and SCF complexes leads to the accumulation of their substrates, resulting in abnormal cell cycle progression. Thus, these complexes are attractive targets for cancer therapy[25,26].

CUL1 and the two almost-identical CUL4A and CUL4B (CUL4) act as molecular scaffolds for their respective CRLs. These cullin scaffolds associate with specific adapters, including either SKP1 or DDB1 (damage-specific DNA-binding protein 1) and RBX1, to recruit E2 ubiquitin ligases[11,27]. Although CRLs share a similar architecture, SCF utilizes F-box proteins to recognize phosphorylated forms of target substrates[28–30], whereas CRL4 requires members of the WD40-domain containing DDB1/CUL4-associated factor (DCAF) protein family as substrate receptors[27,31,32]. For example, CRL4-mediated ubiquitination of the licensing complex member CDT1 requires a DCAF,

CDT2[13,33], which interacts with CUL4 and DDB1 to facilitate the degradation of CDT1 in a CDC48/p97-dependent pathway[34,35]. DCAFs often recognize substrates that contain PCNA (proliferating cell nuclear antigen)-binding motifs (PIP boxes), but CUL4 is also detected on chromatin during the G1 phase of the cell cycle[36], suggesting that it can be recruited to chromatin without PCNA.

The replication origin binding protein RepID (also known as DCAF14, as well as pleckstrin homology domain-interacting protein, or PHIP) is a member of the DCAF family that contains a bromo domain and cryptic tudor domain in addition to the WD40 domain[32,37,38]. RepID is a multifunctional protein that facilitates cell proliferation by inhibiting apoptosis[39–42] and is a biomarker for aggressive metastatic melanoma[43]. Chromatin-bound RepID selectively interacts with a group of replication origins and promotes initiation from these origins, possibly by facilitating the formation of chromatin loops[37]. Abnormal replication patterns in the absence of RepID suggest a role for RepID in establishing and regulating the orderly progression of replication origin activation.

Here, we report that RepID is required for CUL4 recruitment on chromatin during the early stages of the cell cycle. We observed that RepID-depleted cells exhibited diminished CUL4 chromatin loading, resulting in accumulation of CRL4 substrates (e.g. CDT1, p21, and SET8) on chromatin. These molecular events correlated with abnormal cell cycle progression, massive re-replication in the absence of CDT2, marked genomic instability and low sensitivity of RepID-depleted cells to the neddylation inhibitor MLN4924 (pevonedistat). We also observed that the alternative ubiquitin ligase complex, SCF, associated with a distinct group of replication origins that did not bind RepID and partially compensated for the absence of RepID by facilitating CDT1 degradation. RepID depleted cells exhibited remarkably high sensitivity to SKP2 inhibitors. These results suggest that RepID plays a role in ensuring genomic stability by recruiting the critical ubiquitin ligase component CUL4 to chromatin prior to the onset of DNA replication. These observations provide an example for the differential regulation of distinct groups of replication origins by specific protein–DNA interactions.

## Results

**CUL4 is recruited to chromatin through the RepID WD40 domain.** To determine whether RepID plays a role in mediating the chromatin association of the cullin proteins, we measured the levels of chromatin-bound cullins in three cancer cell lines (U2OS, HCT116, and K562) in which we depleted RepID (RepID knock-out [KO]) by CRISPR-cas9 gene editing (Supplementary Fig. 1a). The absence of RepID did not affect the whole-cell expression levels (WCL) of all seven mammalian cullins (Fig. 1a). However, the chromatin-bound (CB) fractions of CUL4A and CUL4B were significantly reduced (more than 80%) in all 3 RepID-depleted cell lines whereas changes in the association of other cullins with chromatin were inconsistent (Fig. 1a, b). Chromatin-bound levels of other components of the CRL4 complex (DDB1, DDB2, CDT2, and RBX1) or PCNA were not consistently altered by RepID depletion (Fig. 1a, b). Depletion of other DCAFs such as DDB2 or CDT2 had no effect on chromatin-bound levels of CUL4 (Supplementary Fig. 1b). These observations suggest that recruitment of CUL4 protein to chromatin requires RepID.

We next determined which domain of RepID was important for CUL4 chromatin recruitment. For this, we generated stable cell lines expressing FLAG-tagged RepID fragments (Fig. 1c) in a RepID KO background and immunoprecipitated nuclear proteins using FLAG-agarose beads. As expected, CUL4 was able to bind

the full-length (FL) RepID (Fig. 1d). The interaction between RepID and CUL4 was validated by immunoprecipitation with antibodies directed against the endogenous CUL4A and CUL4B (Supplementary Fig. 1c). An interaction with CUL4A/B was also evident with the RepIDΔ3 fragment, which contained the WD40 domain, but not with the F2–5 or F3–5 RepID fragments, both lacking the WD40 domain (Fig. 1d). RepID did not interact with the other nuclear cullins (Fig. 1d). In concordance, in vitro binding analyses (Supplementary Fig. 1d) demonstrated that the RepID WD40 domain could bind directly to the CRL4 components CUL4A and DDB1.

CUL4 association with chromatin was significantly higher in cells expressing a WD40-containing RepID fragment (FL or Δ3)

than in cells that were either depleted of RepID or contained RepID fragments without the WD40 domain (Fig. 1e, f). The chromatin associations of the other cullins were not affected by the expression of the full length RepID, nor by the expression of the different fragments of RepID. These results clearly show that CUL4 recruitment to chromatin is mediated by an interaction with the WD40 domain of RepID.

Next, we investigated the distribution of CUL4 on chromatin during the cell cycle. CUL4 was present in all chromatin fractions from elutriated, cell cycle-fractionated RepID-containing cells. Chromatin fractions from RepID-expressing cells contained significantly higher levels of CUL4 than chromatin fractions from RepID-depleted cells (Fig. 1g and Supplementary Fig. 1e),

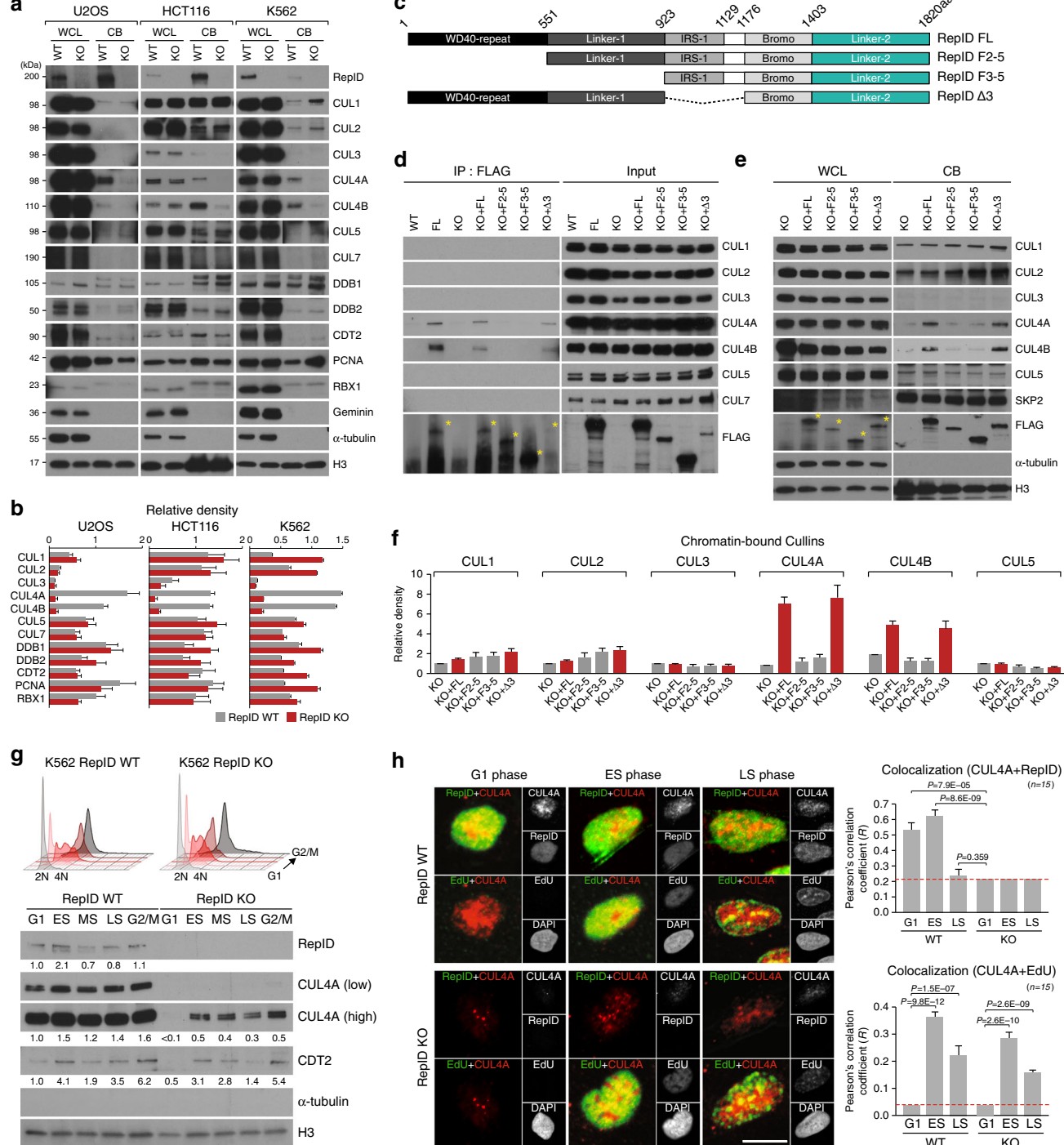

verifying that RepID was essential for CUL4 recruitment to chromatin. The effect of RepID depletion on CUL4 recruitment was strongest in the G1 phase of the cell cycle, as CUL4 did not bind chromatin in G1 fractions of RepID depleted cells (Fig. 1g and Supplementary Fig. 1e). The effects of RepID were specific for CUL4 recruitment, as levels of CDT2, an interacting substrate receptor for CUL4 recruited at the beginning of S-phase, were similar in RepID-expressing and RepID-depleted cells (Fig. 1g). These results were consistent with confocal microscope observations, in which CUL4A was present in pre-extracted nuclei and mostly colocalized with RepID during G1 and early S phase, whereas colocalization was lower in late S-phase, concomitant with the dissociation of RepID from chromatin (Fig. 1h, Supplementary Fig. 1f, g and Supplementary Data 1).

**CUL4 recruitment is required for proper cell cycles**. We next tested if RepID-mediated depletion of CUL4 from chromatin altered cell cycle progression. In RepID-expressing cells, depletion of either CUL4A or CUL4B did not affect the cell cycle distribution (Fig. 2a, b), probably because of a compensatory mechanism whereby silencing of CUL4A led to mildly enhanced recruitment of CUL4B and vice versa. This compensatory effect was also detected in RepID-depleted cells (Fig. 2b). Depletion of both CUL4 variants (CUL4A + B) in cells with intact RepID correlated with a low but consistent increase in the fraction of G2/M cells. In RepID-depleted cells, depletion of CUL4A + B led to an increased G2/M fraction, significantly decreased S-phase EdU incorporation and a markedly increased fraction of re-replicating cells (Fig. 2a), together with accrued γH2AX and phosphorylated replication protein A (RPA), two markers of DNA damage (Fig. 2c). PCNA loading on chromatin was not affected by either RepID or CUL4 status (Fig. 2c), suggesting that decreasing the amount of chromatin-bound CUL4 induces abnormal cell cycle progression without affecting the binding of PCNA to chromatin.

To directly evaluate whether cell cycle progression was affected by RepID depletion, we treated RepID-depleted and RepID-expressing cells with nocodazole to arrest them in metaphase, and then released the cells into nocodazole-free medium. In RepID-expressing cells, 44% of the mitotic cells progressed into the G1 phase within 3 h after release from nocodazole, and a third of the cells started DNA synthesis (S phase) after 6 h (Fig. 2d). A smaller fraction of RepID KO cells progressed from mitosis to G1 phase (15.9% of KO cells after 3 h). Similarly, a smaller fraction of RepID KO cells progressed from G1 to S phase (Fig. 2d, f right panel), and RepID KO cells exhibited a higher fraction of cells that have undergone partial or complete genome re-replication (21.9% of KO cells vs. 6.42% of WT cells after 48 h, Fig. 2d).

RepID expressing cells exhibited rapid degradation of CRL4 substrates after the G1/S transition, but RepID depleted cells with low chromatin-bound CUL4A showed slow degradation of CDT1 and almost no degradation of both CRL4 substrates SET8 and p21 (Fig. 2e and Supplementary Fig. 2a, b), even though a significant fraction of cells were in S-phase 6 h after release (Fig. 2d). The interaction ratios between CDT1 and PCNA or between CDT1 and geminin were not affected by RepID depletion, suggesting that CDT1 degradation was independent of PCNA and geminin (Supplementary Fig. 2c, d). Consistent with the delayed degradation of CRL4 substrates, both γH2AX and 53BP1 accumulated on chromatin with increased micronuclei in RepID KO cells (Fig. 2e and Supplementary Fig. 2e–g), suggesting that RepID KO cells are subject to persistent DNA replication stress. These results clearly show that RepID-dependent CUL4 recruitment on chromatin is crucial for proper cell cycle progression (Fig. 2g).

**Disparate origins bind CUL4A partnered with distinct DCAFs**. Consistent with previous reports[13,15,23], re-replication was also detected in cells in which CDT2 was depleted (Fig. 3a). RepID-depleted cells that were also exposed to CDT2 siRNA exhibited a more severe re-replication phenotype than RepID expressing cells, along with an increased number of non-replicating S-phase cells and apoptotic cells (cells that underwent re-replication: 15.3% in RepID WT + si-CDT2 vs 29.5% in RepID KO + si-CDT2, non-replicating S-phase cells: 2.6% in RepID WT + si-CDT2 vs. 5.3% in RepID KO + si-CDT2, apoptotic cells: 2.8% in RepID WT + si-CDT2 vs. 13.0% in RepID KO + si-CDT2) (Fig. 3a). This increased sensitivity suggests an adaptation to the low abundance of CUL4 on chromatin. In aggregate, our observations suggest that proper CUL4 recruitment to chromatin requires RepID, but efficient prevention of re-replication requires both DCAFs (RepID and CDT2).

Our previous studies have shown that RepID associates with a subset of replication origins. To determine whether CRL4 was also associated with replication origins, we performed chromatin immunoprecipitation followed by sequencing (ChIP-Seq) using anti-CUL4A, anti-CDT2 and anti-FLAG antibodies to pull-down FLAG-tagged RepID in U2OS WT cells. Of the total CUL4A peaks (56,715), 77.2% (43,812) were associated with replication origins (Fig. 3b). Conversely, 37.5% of origins (47,810 of 127,325) were associated with CUL4A peaks. Approximately 30% of CUL4A peaks colocalized with CDT2 peaks (Fig. 3b—groups 3 and 4, 16,883 peaks). RepID associated with 15.5% of CUL4A peaks (Fig. 3b—groups 2 and 4,8,815 peaks). Interestingly, only 3.9% (group 4,2,226 peaks) of CUL4A peaks were associated with

**Fig. 1** RepID is required for CUL4 loading on chromatin. **a** Levels of RepID, cullins and their complexes from wild-type (WT) or RepID-depleted (KO) U2OS, HCT116 and K562 cancer cell lines. Histone H3 and α-tubulin were used as loading controls. WCL, whole cell lysates; CB, chromatin-bound proteins. **b** Quantification of relative intensities of chromatin-bound protein signals analyzed as shown in **a** after normalization with respect to histone H3 signal intensities in each cell line. CUL4 was indicated as bold. Error bars represent standard deviations from three independent experiments. **c** Construction of RepID fragments. **d** Soluble nuclear and chromatin-bound fractions were extracted from U2OS cells transfected with RepID fragments as indicated. A pull-down assay using a FLAG antibody was performed and co-precipitated cullins were analyzed by immunoblotting. Yellow asterisks, FLAG-RepID fragments. **e** Cullin levels in WCL and CB from RepID KO U2OS cells transfected with RepID constructs as indicated. **f** Quantification of chromatin-bound cullin levels, normalized to chromatin bound levels in RepID KO cells. Error bars represent standard deviations from three independent experiments. **g** K562 RepID WT and KO cells were fractionated by elutriation and cell cycle status of each fraction (G1, early [ES], middle [MS], and late [LS] S-phase and G2/M) was confirmed by FACS (upper panel). Chromatin-bound proteins were analyzed by immunoblotting (bottom panel). The numbers under the panels represent the intensity ratios for each protein normalized by the intensity of the signal at G1 phase in RepID WT from three independent experiments (bold). **h** U2OS cells were EdU-labeled, pre-extracted and chromatin-bound RepID (green), chromatin-bound CUL4A (red), DNA content (DAPI) and EdU (magenta) were detected. Cells in ES and LS were identified by EdU staining patterns and G1 and G2 nuclei (EdU negative) were identified by DAPI intensity. To clearly observe colocalization with EdU and CUL4A, magenta was converted to green (left panel). The extent of colocalization between RepID and CUL4A, or between EdU and CUL4A (right panel). Pearson's correlation coefficients (n = 15) and p-values were calculated using a two-tailed t-test. RepID KO cells were used to estimate background (red line, right panel). Scale bar indicates 10 μm

both CDT2 and RepID, suggesting that CUL4A-RepID and CUL4A-CDT2 complexes may bind distinct groups of origins and that the identity of the CRL4-associated DCAF may have a regulatory significance.

To compare directly the extent of colocalization between replication origins, RepID binding regions and other chromatin features (e.g., histone modifications), we quantified the association between the various groups and sequences associated with chromatin modifications that mark early and late replicating DNA. As shown in Fig. 3c and Supplementary Fig. 3a, for each

association we calculated the above median integral (AMI), a quantifier of peak colocalization that takes data variance into account[44] (http://discovery.nci.nih.gov/Coloweb/ and http://projects.insilico.us.com/ColoWeb/). Interestingly, origins associated with both RepID and CUL4A were more likely to colocalize with H3K4me3, a hallmark of early replicating origins, than with H3K9me3, a hallmark of late replicating origins[45] (Fig. 3c, d; in group 2, AMIs were 2171.5 with H3K4me3 vs. 67.4 with H3K9me3; in group 4, AMIs were 1991.4 with H3K4me3 vs. 465.5 with H3K9me3). CUL4A peaks that were not associated with

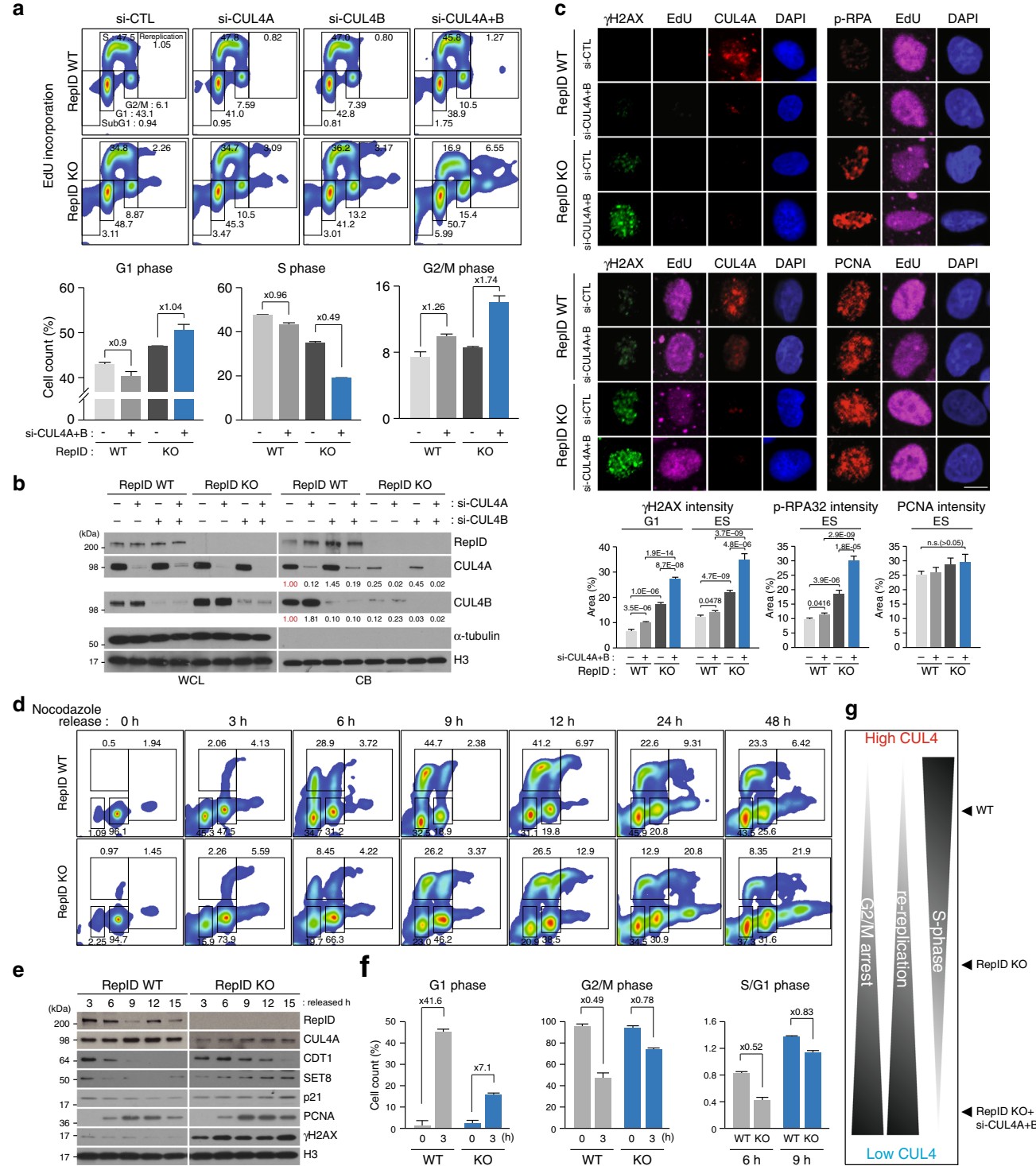

RepID (group 1 and 3) were enriched with H3K9me3 peaks and depleted in H3K4me3 peaks. Acetylated histones (H3K9Ac and H3K27Ac), which are known as early replication marks, showed higher colocalization with RepID peaks, including CUL4A peaks colocalizing with RepID, than with CUL4A peaks colocalizing with

CDT2 (Supplementary Fig. 3a, compare groups 2, 4, and 6 with groups 1, 3, and 5). In concordance, RepID binding sites preferentially colocalized with early replicating origins, whereas CDT2 binding did not exhibit particular replication timing preference (Supplementary Fig. 3b). In aggregate, these results

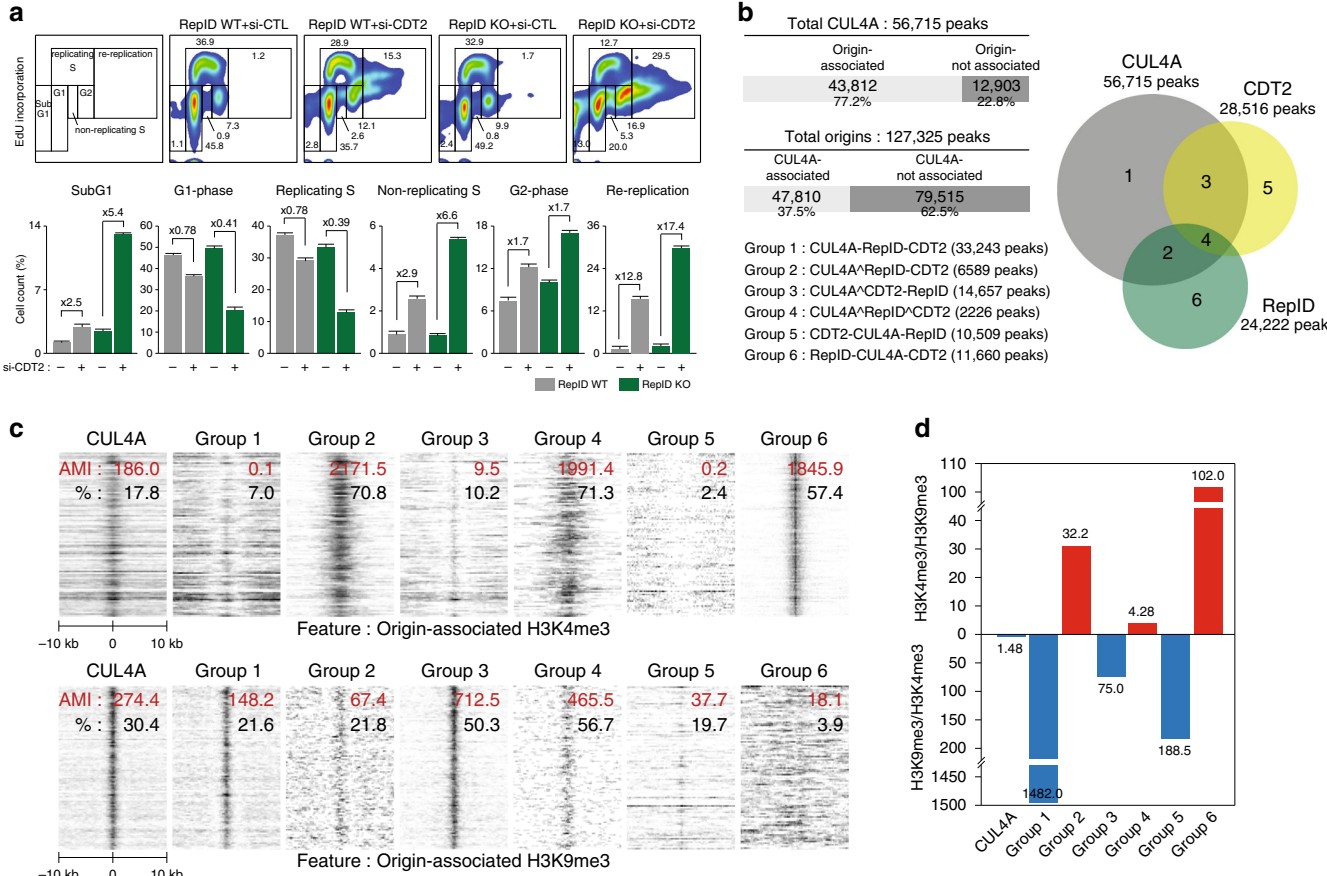

**Fig. 3** CUL4A colocalizes with distinct replication origins and different DCAFs during S phase. **a** CDT2-siRNA transfected RepID WT and KO U2OS cells were labeled with EdU for 30 min and analyzed by flow cytometry. Percentages of cells in each cell cycle phase are indicated in the flow cytometry plots (upper panel) and histograms (bottom panel). Fold changes were based on the values from control-siRNA transfected cells. Data are representatives of three independent experiments. **b** Left, table indicating colocalizations between CUL4A ChIP-Seq and replication origins (^: intersection, -: subtraction). Right, Venn diagram comparing CUL4A, CDT2 and RepID ChIP-Seq results. Three subgroups indicate CUL4A binding sites overlapping with RepID (Group 2), CDT2 (group 3), RepID and CDT2 (group 4) while three subgroups represent exclusive binding sites for CUL4A (group 1), CDT2 (group 5) and RepID (group 6). **c** Heat maps showing colocalization between the six subgroups defined in **b** and sites enriched in methylated histone H3 (H3K4me3, top panel and H3K9me3, bottom panel). AMI value (red) and colocalized percentage (black) are indicated. Twenty kb windows were used for the analysis. **d** A centered bar chart representing the AMI ratio from **c** between H3K4me3 and H3K9me3

**Fig. 2** RepID-dependent CUL4 recruitment on chromatin is required for cell cycle progression. **a** siRNAs for CUL4A, CUL4B or CUL4A + B were transfected into RepID WT and KO U2OS cells. After 5 days, cells were labeled by EdU for 30 min, collected and analyzed by flow cytometry. Percentages of cells in each cell cycle phase are indicated (upper panel). Graphs depicting the distributions of G1, S and G2/M phase cells in control vs. CUL4A + B siRNA transfected cells are shown in the bottom panel. Error bars represent standard deviations from three independent experiments. **b** Chromatin-bound proteins were isolated from cells collected as in **a** and analyzed by immunoblotting. The numbers under the panels represent the intensity ratios compared to the intensity of the signal in control-siRNA transfected RepID WT cells. **c** CUL4A + B-siRNA transfected RepID WT and KO U2OS cells were labeled with EdU for 30 min, and immunofluorescence analysis was performed using CUL4A, γH2AX, phospho-RPA, or PCNA antibodies after pre-extraction (upper and middle panel). Scale bar indicates 10 μm. The intensity of each protein was determined in G1 (EdU negative) and ES (EdU positive) using the Pearson's correlation coefficient. *p* values were calculated using a two-tailed *t*-test (*n* = 10) (bottom panel). **d** Mitotic HCT116 RepID WT or KO cells were collected after released from a nocodazole block, reseeded in fresh medium and harvested at the indicated time points after incubation with EdU for 30 min. Cells were then analyzed by flow cytometry, and the percentage of cells in each cell cycle phase is indicated as in **a**. **e** Aliquots of cells in **d** were analyzed by immunoblotting after extraction of chromatin fractions. **f** Graphs representing the percentages of cells in G1, G2/M phase and the ratio of cells

suggest that RepID associates with early replicating origins and recruits CUL4 to those origins. These observations support the notion that the association of CUL4A/B with distinct groups of replication origins depended on interactions with specific DCAFs, and that this differential association was linked to replication timing.

**RepID increases the sensitivity of cancer cells to MLN4924.** Neddylation, the addition of NEDD8 to CRL4, is essential for the activity of the CRL4 complex[46,47]. We examined the sensitivity of both RepID WT and RepID KO cells to MLN4924 (pevonedistat), a selective NEDD8 activating enzyme inhibitor[46]. Both RepID-expressing cells and RepID depleted cells exposed for 2 days to MLN4924 exhibit a fraction of cells that re-replicate genomic DNA (EdU incorporation with DNA content higher than 4N) with RepID expressing cells showing a somewhat higher fraction of re-replicating cells than RepID-depleted cells (250 nM: 15.8% vs. 11.9 % in WT and KO, respectively; 500 nM: 57.5% vs. 37.7% in WT and KO) (Fig. 4a, b). Levels of CRL4 substrates including CDT1, SET8, and p21 were clearly higher in RepID WT cells than in RepID KO cells following MLN4924 treatment (Fig. 4c).

The sensitivity of RepID KO cells to MLN4924 increased when these cells were transfected with the RepID FL plasmid (64% re-replicating cells) or the plasmid Δ3 containing WD40 domain (60% re-replicating cells) but not with the F2-5 fragment lacking

the WD40 domain (37.3% re-replicating cells) (Fig. 4d, e and Supplementary Fig. 4). We also measured the levels of chromatin-bound CDT1 to test the hypothesis that re-replication is mediated by the persistence of CDT1 on chromatin. EdU-labeled nuclei were pre-extracted before fixation to remove nuclear soluble proteins[48], followed by incubation with CDT1 and FITC-conjugated secondary antibodies. Consistent with the above results, 70.1% of chromatin-bound CDT1 was detected in RepID KO cells that had undergone re-replication, whereas 89.0% was in re-replicating fractions in RepID WT cells (Fig. 4d, e). These results suggest that reduced CUL4A-chromatin binding in the absence of RepID decreases the sensitivity of cells to MLN4924.

**SKP2 inhibitors synergize with RepID depletion.** The alternative ubiquitin ligase complex, SCF, also ubiquitinates CDT1 in S and G2 phases[49]. We asked whether SCF compensates for defective CDT1 degradation by CRL4 in RepID KO cells. We first tested which cullins were recruited to chromatin at different stages of the cell cycle. Levels of chromatin-bound CUL1 were significantly increased in RepID KO cells, particularly in late S-phase, 9–15 h after release from nocodazole (Fig. 5a, b), consistent with the timing of CDT1 degradation (Fig. 2e). In addition, interactions between CUL1 and CDT1 appeared during middle and late S-phase, and such interactions were higher in RepID KO cells than in RepID WT cells (Supplementary Fig. 5).

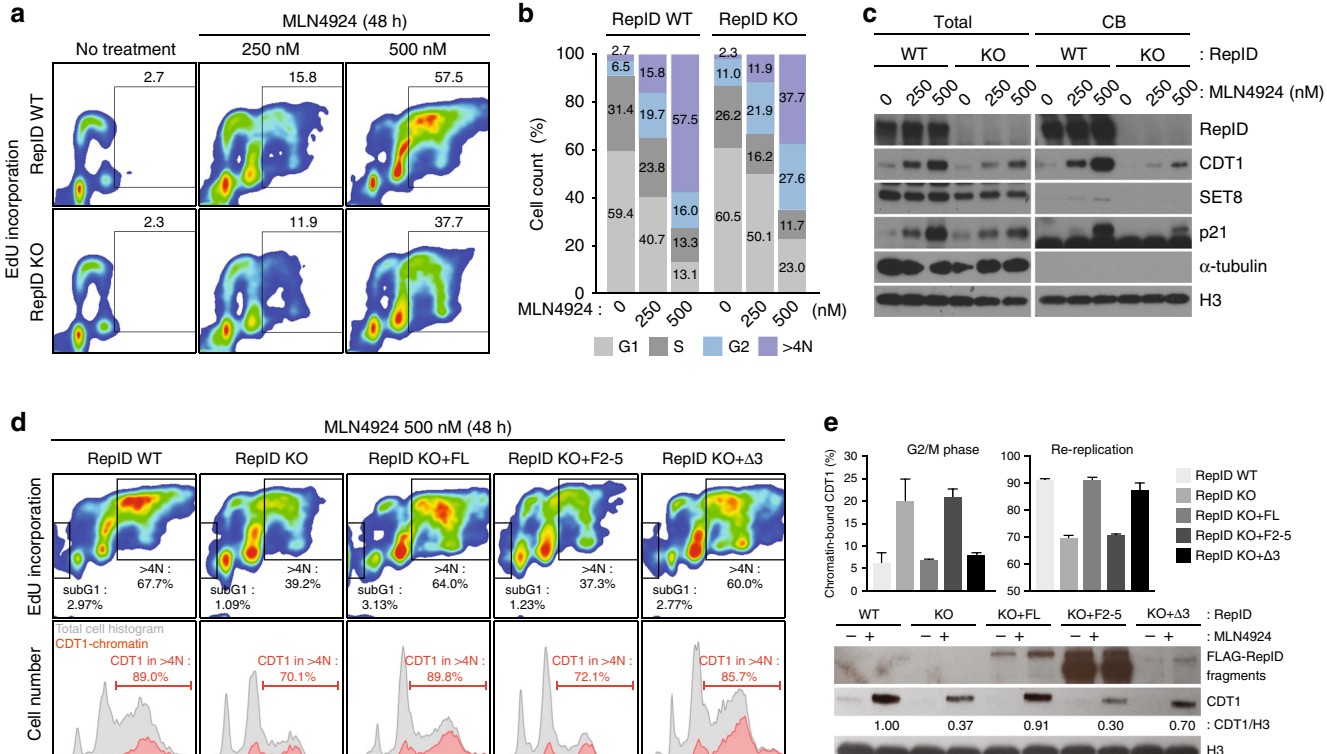

**Fig. 4** RepID depletion confers resistance to the neddylation inhibitor MLN4924. **a** U2OS WT or RepID KO cells were incubated with or without 250 and 500 nM MLN4924 for 2 days. Cells were labeled with EdU for 30 min prior to collection and analyzed by FACS. Percentage of re-replicating cells is indicated. **b** Bar chart depicting the cell cycle distribution of cells collected in **a**. **c** Chromatin-bound (CB) proteins isolated as in **a** and analyzed by immunoblotting using antibodies directed against CRL4 substrates including CDT1, SET8, and p21. **d** WT, RepID KO U2OS cells and RepID KO U2OS cells stably transfected with RepID variants were treated with 500 nM MLN4924 for 2 days and labeled with EdU for 30 min prior to collection and analyzed by FACS. Percentage of subG1 and re-replicating cells are indicated (upper panels). Percentages of chromatin-bound CDT1 in re-replicating cells are indicated (red, bottom panel). **e** Graphs depicting the percentage of chromatin-bound CDT1 in cells in G2/M phase (left panel) or in re-replicating cells (right panel). The intensity of chromatin-bound CDT1 in specific cell cycle fractionated cells isolated as in **d** was measured by immunoblotting (bottom panel). The numbers under the panel represent the ratios of CDT1 intensities relative to the intensity of the signal in MLN4924-treated RepID WT cells after normalization with histone H3 from three independent experiments

Increased levels of chromatin-bound SKP2 were also observed with kinetics similar to CUL1 levels (Fig. 5a, b), and RepID KO cells that have reached the second S-phase after the initial release seemed to have adapted to the absence of RepID by recruiting higher levels of SKP2 and CUL1 on chromatin.

To determine whether SCF also associated with replication origins, we performed ChIP-Seq using anti-SKP2 in HCT116 RepID WT and KO cells. In HCT116 WT cells, 50.9% of SKP2 peaks (6216 of 12,221 peaks) were associated with replication origins (Fig. 5c). Conversely, 8.6% of origins (6759 peaks) were associated with SKP2 peaks. SKP2 also colocalized with origins in

RepID KO cells (53.4%, 7409 peaks), and 9.3% of origins (7965 peaks) were associated with SKP2. However, in contrast with RepID associated origins, SKP2 associated origins were more likely to colocalize with H3K9me3, and this association was stronger in RepID-deficient cells than in RepID WT cells (AMI values: 327.7 in RepID KO vs 184.7 in RepID WT) (Fig. 5d, e). These observations suggest that SKP2 binds replication origins associated with histone marks of late-replication chromatin, consistent with increased SKP2 binding on chromatin in late S phase (Fig. 5a).

We have not observed significant colocalization between RepID associated binding sites and SKP2 binding sites, or

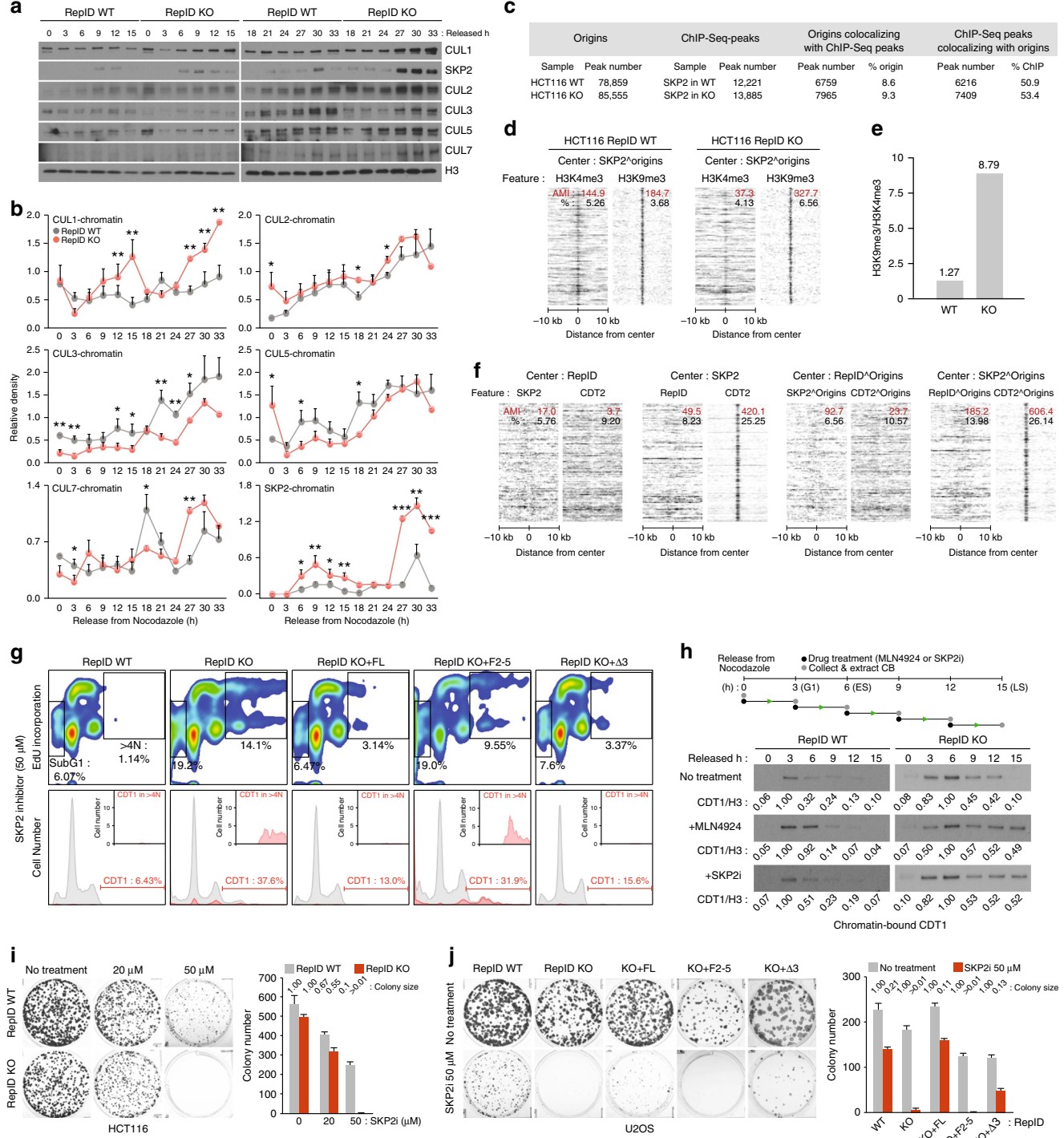

between RepID binding sites and CDT2 binding sites (Fig. 5f, left panel, center: RepID). However, SKP2 binding sites highly colocalized with CDT2 binding sites (Fig. 5f, second panel, center: SKP2). Similarly, RepID associated origins did not colocalize with CDT2 associated origins or with SKP2 associated origins whereas SKP2 associated origins colocalized with CDT2 associated origins (Fig. 5f, right panels, center: RepID^Origins and SKP2^Origins). These results suggest that the SKP2-containing complex, SCF, might compensate for lack of CRL4 function in RepID KO cells by binding origins that are not associated with RepID during the late S-phase. Because RepID associated origins are dysfunctional in RepID KO cells, SKP2 binding might be essential for proper progression of DNA replication in the absence of RepID.

To determine if SCF-mediated ubiquitination is critical for S-phase progression in cells that are deficient in RepID, and thus contain low levels of chromatin-bound CRL4, both RepID WT and RepID KO cells were exposed to the SKP2 inhibitor (SKP2i) SZL-P1-41[25]. As shown in Fig. 5g, exposure to 50 μM SKP2i resulted in a marked increase in the fraction of re-replicating cells among cells deficient in RepID compared with cells with intact RepID (14.1% in RepID KO vs. 1.14% in WT cells). The fraction of apoptotic cells also increased in RepID-deficient cells (19.2% in RepID KO vs. 6.07% in WT). Increased re-replication in SKP2i-treated RepID KO cells was associated with higher levels of chromatin-bound CDT1 (37.6% in RepID KO vs. 6.43% in WT) (Fig. 5g, bottom panel). Re-introduction of the RepID FL or RepID fragments containing the WD40 domain in RepID KO cells restored both the re-replication rates and CDT1 induction to levels similar to those of control cells (Fig. 5g, RepID KO + FL and KO + Δ3 panel). In contrast, F2–5 RepID rescued cells (lacking the WD40 domain) were highly sensitive to SKP2i with similar rates of re-replication and CDT1 induction as those of RepID KO cells.

We then determined the effects of NEDD8 and SKP2 inhibitors on the activity of SCF and CRL4 by directly measuring the levels of chromatin-bound CDT1. WT and RepID KO cells synchronized by nocodazole were released with fresh media and treated with MLN4924 or SKP2i for 3 h before collection (Fig. 5h). In cells containing intact RepID, CDT1 degradation at the G1-S boundary was abolished by MLN4924, but not by SKP2i (Fig. 5h, left panel), suggesting that CRL4 is a major factor for CDT1 degradation in those cells. In contrast, CDT1 degradation during late S phase was prevented by both drugs in RepID-deficient cells (Fig. 5h, right panels), suggesting that CDT1 degradation was catalyzed by SCF. Overall, these combined data suggest that CRL4 and SCF act sequentially during the cell cycle. In agreement, SKP2 inhibition had no effect in RepID-expressing cells, probably because CRL4 is the main player in CDT1 degradation. However, SCF-mediated

CDT1 ubiquitination becomes critical in RepID KO cells, where the CRL4 complex is inactive on chromatin. Consistent with the above, cells expressing intact RepID or the WD40-containing RepID fragment (KO + FL or KO + Δ3) were resistant to SKP2i, whereas RepID KO or KO + F2–5 cells showed high sensitivity to SKP2 inhibition with no surviving colonies observed after 50 μM SKP2i treatment (Fig. 5i, j). These results imply that cancer cells with defective CUL4 and/or RepID protein may be more sensitive to SKP2 inhibition. Taken together, our observations suggests that RepID expression levels might modulate the sensitivity of cancer cells to cullin-targeting drugs.

## Discussion

The data presented here demonstrate that RepID is crucial for recruiting CUL4 to chromatin and for facilitating proper DNA replication and damage responses by mediating the ubiquitination of replication-associated substrates. Of those substrates, CDT1 is required for origin licensing and mitosis[50,51], and SET8 degradation is essential for proper S-phase progression and prevention of G2/M checkpoint activation, spontaneous DNA damage and re-replication[20,52,53]. P21 also exhibits a similar pattern of CRL4-mediated degradation as CDT1, leading to the activation of the G1/S transition and the G2/M checkpoint when CUL4 was depleted[22–24,26]. Unlike other DCAFs, RepID interacts directly with chromatin via its bromodomain and cryptic Tudor domain[38,54], whereas other DCAFs such as CDT2 need PCNA as a platform to associate with chromatin[16,21,22,33,55,56]. Our results show that CUL4 binds chromatin during G1, when PCNA is absent, and that levels of chromatin-bound CUL4 are reduced in RepID depleted cells during G1 as well as during S-phase. These observations suggest that RepID mediates the recruitment of CUL4 in PCNA-free G1 phase cells, enabling subsequent CDT2-mediated degradation of CDT1 during the S-phase. The residual low levels of CUL4 detected on chromatin in RepID KO cells might be due to interactions with other proteins. As S-phase progresses, our data suggest that CRL4 components, including RepID, gradually leave chromatin. The ubiquitination of remaining CRL4 subtrates, including CDT1, is mediated by the SKP2-containing SCF complex. In concordance, RepID KO cells exhibit a striking sensitivity to SKP2 inhibition. Failure to ubiquitinate CDT1 due to RepID depletion induces partial genome re-replication and a marked increase in genomic instability, which is strongly enhanced when both RepID and SKP2 are depleted.

We observed that CUL4A associates with a large cohort of replication origins, whereas both DCAFs (CDT2 and RepID) and the F-box protein SKP2 associate each with a smaller, distinct sub-group of origins. Previous reports have shown that CUL4A

**Fig. 5** SKP2 inhibitors synergize with RepID depletion to kill cancer cells. **a** HCT116 cells released from nocodazole were collected every 3 h for up to 33 h, and chromatin-bound cullins were analyzed using immunoblotting. **b** Graphs showing relative intensities of cullins and SKP2 calculated from blots derived from (a) and normalized using histone H3 levels. Error bars represent standard deviation from three independent experiments (*p-value < 0.05, **p < 0.01, ***p < 0.001, no asterisk: n.s.). **c** Table depicting the colocalization between SKP2 ChIP-seq peaks and replication origins in HCT116 WT or RepID KO cells. **d** Heat map showing the colocalizations between origin-associated SKP2 and the H3K4me3 or H3K9me3 peaks (AMI, red; colocalized percentage, black; ^, intersection). **e** Bar graph representing the AMI ratios from **d**. **f** Heat maps for colocalizations between RepID and SKP2 or CDT2 peaks (first panel), between SKP2 and RepID or CDT2 peaks (second panel), between the origin-associated RepID and the origin-associated SKP2 or origin-associated CDT2 (third panel), and between the origin-associated SKP2 and origin-associated RepID or origin-associated CDT2 (fourth panel). window scale, 20 kb. **g** WT, RepID KO U2OS cells and RepID KO U2OS cells transfected with the specified RepID fragments were treated with 50 μM SKP2 inhibitor for 2 days and labeled with EdU. Cell cycle (upper panel) and chromatin-bound CDT1 (bottom panel) were analyzed as in Fig. 4d. **h** Top, a schematic of the experimental procedure. Mitotic HCT116 WT or RepID KO cells were released in fresh medium and collected every 3 h with MLN4924 or SKP2 inhibitors added during the last 3 h prior to collection. The time zero samples were mitotic cells without drug exposure. Chromatin-bound CDT1 was analyzed and normalized using histone H3. Fold changes were calculated using the highest CDT1 intensity (bold) in each group from three independent experiments. **i**, **j** Colony assay with HCT116 cells (**i**) and U2OS cells (**j**). bar charts, colony number. Ratios for the average size of colonies in each group relative to the size of colonies in untreated cells are provided above the bars. Error bars represent standard deviations from three independent experiments

inactivation severely reduced heterochromatin formation because of the inhibition of the interactions between CUL4A and EED (a subunit of the PRC2 methyltransferase) or JARID1C (H3K4me3 demethylase)[57,58]. We observed that RepID binding sites are enriched in H3K4me3-containing, H3K9Ac-containing, and H3K27Ac-containing chromatin whereas CDT2 binding sites colocalize with H3K9me3 containing chromatin. These observations suggest that RepID-recruited CUL4A binds preferentially to early, euchromatin-associated replication origins.

SCF complex recruitment to chromatin is enhanced in RepID-depleted cells, the SCF component SKP2 binds to replication origins that are not associated with RepID, and cells that are deficient in RepID are hypersensitive to SKP2 inhibitors that affect SCF function. These observations are in line with previous reports demonstrating a functional redundancy between CRL4 and SCF for their common substrate degradation[14], and with studies suggesting that two mechanisms are involved in CDT1 degradation[59]. Interestingly, we observed only a modest increase in CDT1 and SET8 levels in asynchronous RepID depleted cells, consistent with the suggestion that some of the ubiquitination reactions normally performed by CRL4 in RepID proficient cells are compensated by SCF in RepID deficient cells. It is also possible that some RepID KO cells with persistent CDT1/SET8 accumulation have a lower threshold for apoptosis, and these cells are eliminated from the population. These studies are concordant with the low colony formation, DNA damage and increased subG1 populations we observed in RepID KO.

We propose the following model for RepID's role in regulating CRL4 activity and RepID-independent SCF functions during replication (Fig. 6). The CRL4 anchor CUL4 is recruited to chromatin via the architectural DCAF RepID during the G1 phase and associates primarily with early replication origins. During S-phase, the CUL4-containing, chromatin-bound CRL4 associates with PCNA and with the alternative DCAF CDT2 to induce CDT1 degradation. Later in S-phase, the SKP2-containing SCF complex degrades the remaining substrates on late replication origins. In RepID-deficient cells, CUL4 loading to chromatin is compromised, and CDT1 degradation is mediated by SCF. Because the SCF complex acts as the main CRL for substrate degradation in RepID deficient cells, SKP2 inhibitors exhibit increased potency in these cells. We propose that RepID levels might modulate the sensitivity of cancer cells to inhibitors affecting ubiquitin-mediated degradation pathways.

Many cancer types harbor alterations in components of CRL complexes. For example, CUL4A is highly expressed in many types of cancer, and it increases the survival of melanoma cells exposed to UV radiation by promoting nucleotide excision repair[60,61]. RepID expression also promotes melanoma metastasis by regulating the expression of upstream mediators of the IGF axis and downstream mediators of tumor cell invasion[43]. Patients carrying RepID and CUL4B mutations show overlapping phenotypes during development[62], suggesting that both proteins act in the same pathway. These observations lend support to the results presented here, suggesting a novel role for RepID in recruiting CUL4 to chromatin.

Our study has significant clinical implications. Cullin-Ring ligases are now regarded as attractive anticancer targets[46,47]. Components of the CRL4 pathway are known to modulate chemosensitivity in cancer cells, as CUL4 and DDB1 deficiencies exhibit synergistic lethality with camptothecin and UV irradiation[61,63]. Pevenodistat (MLN4924), a selective NEDD8 inhibitor that targets CRL activity, is currently being tested in several Phase I and Phase II clinical trials[64,65] and a SKP2 inhibitor was shown to exhibit antitumor activity in animal models[25]. Our findings uncover important interactions that underlie the

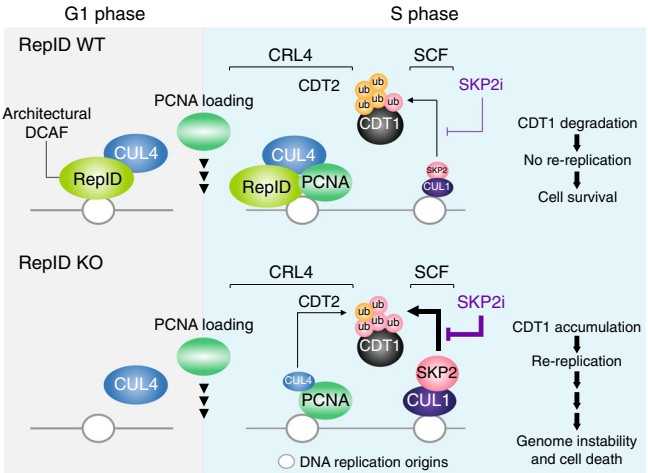

**Fig. 6** Model depicting RepID roles in regulating CRL4 activity and RepID-independent SCF functions during replication. In RepID wild-type (RepID WT) cells, CUL4 is recruited to chromatin by interacting with RepID (architectural DCAF) during G1 in the PCNA-free stage. During S-phase, PCNA is loaded and the CRL4-PCNA complex induces CDT1 ubiquitination using CDT2 (functional DCAF) by associating with early replication origins. Additional CUL4 can be recruited by PCNA during the G1/S transition and early S-phase. The CRL4-PCNA complex associates with CDT2 to degrade CDT1 later during S-phase. SCF is also recruited to prevent CDT1 accumulation at late replication origins. In cells expressing RepID, the SKP2 inhibitor has little effect because of the major role of CRL4 in degrading CDT1. In RepID depleted (RepID KO) cells, CUL4 loading to chromatin is compromised because RepID is not available to recruit it to chromatin during the G1 phase. During S-phase, some PCNA-dependent CUL4 is recruited, but CDT1 degradation is reduced because of low chromatin-bound CUL4 levels. Thus, in RepID KO cells, SCF recruitment to late replication origins compensates for the lack of CRL4 and have a greater role in targeting CDT1. Because SCF now acts as the major ubiquitin machinery for CDT1 degradation, the SKP2 inhibitor shows increased potency, and treatment leads to increased CDT1 levels and re-replication-mediated cancer cell death

dynamic cell cycle effects of CRL4 and SCF in cancer, laying the foundation for new strategies to target altered CRLs in tumors.

## Methods

**Cell culture, chemicals, and synchronization**. Human U2OS, HCT116, and K562 cells with and without RepID deletion[37] were cultured in Dulbecco's modified Eagle's medium (Invitrogen, 10569-010) supplemented with 10% heat-inactivated fetal bovine serum in a 37 °C/5% CO$_2$ humidified incubator. All original cancer cell lines were obtained from ATCC (www.atcc.org). All cell lines were tested for mycoplasmas (Lonza, LT07-418). MLN4924 and SKP2 inhibitors were purchased from Cayman Chemicals (Cat. 15217-1) and Millipore (Cat. 506305), respectively. Drugs were added to the media at the indicated concentrations. HCT116 cells were synchronized in mitosis by a shake-off after 16 h of incubation in 100 nM nocodazole. Mitotic cells were washed three times in phosphate-buffered saline (PBS) and either collected immediately (0 h) or released in fresh medium for different time periods. The cell cycle distribution of cells was confirmed by flow cytometry using LSRFortessa cell analyzer (BD Biosciences) after staining DNA with DAPI.

**RepID depleted cell lines and siRNA transfection**. RepID was depleted in U2OS, HCT116 and K562 cells using CRISPR-CAS9. Twenty base pair guide sequence targeting the fifth exon of RepID (5′-CTGCAAATATGTCATCGACTAGG-3′) for K562 cells and eighth exon of RepID (5′-GTGATAAAATGATCCGAGTCTGG-3′) for U2OS and HCT116 cells were selected from a published database of predicted high specificity protospacer-PAM target sites in the human exome. K562, HCT116, and U2OS cells were cultured in six-well dishes to 70–80% confluency for co-transfection with 2 μg of RepID single guide RNA (sgRNA) plasmid, 2 μg of linearized pCR2.1 vector harboring a puromycin-resistance gene expression cassette, and 10 μl of Lipofectamine 2000 (Life Technologies) per well. Cloning, selection and verification using PCR were performed[37,66]. All SMARTpool

ON-TARGETplus siRNA was purchased from Dharmacon; CUL4A (L-012610), CUL4B (L-017965), DDB2 (L-011022), CDT2 (L-020543), and negative control siRNA (D-001810). siRNA transfection was performed using Lipofectamine RNAiMax (Invitrogen, 13778030).

**Clonogenic survival assay.** Cells were plated in 6-well plates (500 cells/well) in triplicate and treated with MLN4924 or SKP2 inhibitors for 2 weeks. Drug-containing fresh medium was replaced every 2 days. Colonies were fixed, stained with crystal violet, and counted using ImageJ software.

**FACS analysis.** Cells were pulse-labeled with 10 μM EdU for 30 min prior to harvesting and EdU staining using the Click-iT EdU kit (Invitrogen, C10424). Staining was performed according to the manufacturer's protocol. DAPI was used for DNA counterstaining. For chromatin-bound CDT1 analysis, EdU-labeled cells were pre-extracted, fixed and incubated with anti-CDT1 antibodies (Cell Signaling, 8064S) followed by secondary antibodies conjugated with FITC. LSRFortessa cell analyzer (BD Biosciences) with the FlowJo 10.2 software was used for cell cycle analyses.

**Immunofluorescence analysis.** U2OS cells were labeled with EdU for 30 min, rinsed in phosphate buffered saline (PBS) pH 7.4 and then pre-extracted in PBS-T buffer (0.2% Triton X-100 in 1× PBS, phenylmethylsulphonylfluoride [PMSF], protease inhibitor cocktail [Sigma, P8340] and phosphatase inhibitor cocktail [Roche, P4906845001]) for 5 min on ice, followed by 2.0% paraformaldehyde (PFA). Primary antibody staining was performed as follows: anti-FLAG (1:1000, Sigma, F1804), anti-CUL4A (1:200, Abcam, ab92554), anti-γH2AX (1:500, Millipore, 05-636), anti-pRPA (1:500, Bethyl labs, A300-245A) and anti-PCNA (1:200, Santa Cruz, sc-7907). Secondary antibody staining was performed as follows: Alexa 488 conjugated anti-mouse IgG and Alexa 555 conjugated anti-rabbit IgG (1:500, Thermo Fisher Scientific, A11029 and A21428). EdU was detected by Click-iT assay kit according to the manufacturer's protocol. A Zeiss LSM710 confocal microscope was used and Pearson's correlation coefficient (R, or Rcoloc) that is the covariance of the two variables divided by the product of their standard deviations was used for colocalization analysis and calculated using the colocalization Plugin of the FIJI-ImageJ software (https://imagej.nih.gov/ij/index.html). Detailed results of the colocalization analyses are presented in Supplementary Dataset.

**ChIP and ChIP-Seq analyses.** Chromatin immunoprecipitation (ChIP) analyses were performed with 1% formaldehyde-fixed U2OS or HCT116 cells using the Millipore ChIP assay kit (Cat. No. 17-295). Antibodies included normal rabbit IgG (sc-2027), anti-CUL4A (Abcam, ab92554), anti-CDT2 (Abcam, ab72264), anti-SKP2 (Abcam, ab68455), anti-H3K4me3 (Cell Signaling, 9751P) and anti-H3K9me3 (Cell Signaling, 13969P). For ChIP-Seq experiments, immunoprecipitated chromatin was sequenced and peaks were called against IgG controls by SICER using the Genomatix suite (https://www.genomatix.de/) and using a window size of 200 bp, gap size of 600 bp and false discovery rates (FDRs) of 0.01 for the histone marks and 0.00001 for CUL4A (p-value = 0.05). Pull-downed chromatin by CDT2 and SKP2 was sequenced and peaks were called against IgG controls by MACS2 broad peaks program. Overall, 56,715, 28,516, 12,221, 25,181, and 57,632 regions were enriched for CUL4A, CDT2, SKP2, H3K4me3, or H3K9me3, respectively.

**Nascent strand abundance assay and colocalization analyses.** Replication origins were identified using the nascent-strand sequencing and abundance assay[67]. Briefly, DNA fractions (0.5–2 kb) were isolated using a sucrose gradient. Five prime single-strand DNA ends were phosphorylated by T4 polynucleotide kinase (NEB, M0201S) and then treated with lambda-exonuclease (NEB, M0262S) to remove genomic DNA fragments that lacked the phosphorylated RNA primer. Single stranded nascent strands were random-primed using the Klenow and DNA Prime Labeling System (Invitrogen, 18187013). Double-stranded nascent DNA (1 μg) was sequenced using the Genome Analyzer II (Illumina).

Origin peaks were called against untreated total genomic DNA controls as described[45]. Peaks were called against genome controls by SICER using a window size of 200 bp, gap size of 600 bp and false discovery rates (FDRs) of 0.01. Colocalization analyses comparing the locations between replication origins, between origins and protein-bound sequences or chromatin modifications were performed with the java scripts BedIntersect and BedSubtract[66] using the web-based ColoWeb program (http://discovery.nci.nih.gov/Coloweb/; http://projects.insilico.us.com/ColoWeb/) and the Genomatix suite. Colocalization of genomic loci were investigated with the Integrative Genomics Viewer (IGV) (http://www.broadinstitute.org/igv/).

**Chromatin fractionation, co-IP and Western blotting.** Cells were harvested and incubated in cytosol extraction buffer containing NP-40 (20 mM Tris-HCl pH 7.4, 10 mM NaCl, 3 mM MgCl₂, 0.5% NP-40, PMSF, protease inhibitor cocktail and phosphatase inhibitor cocktail). Cells were harvested by centrifugation at 2700×g for 5 min at 4 °C, washed and resuspended in nucleus extraction buffer (10 mM

Tris-HCl pH 7.4, 100 mM NaCl, 1.0% Triton X-100, 1 mM EDTA pH 8.0, 1 mM EGTA, 0.1% SDS, 10% glycerol, 0.5% sodium deoxycholate, protease inhibitor cocktail and phosphatase inhibitor cocktail). The suspension was vortexed, incubated on ice, and then centrifuged at 5200×g for 5 min at 4 °C. The pellet was resuspended with nucleus extraction buffer containing 5 mM CaCl₂ and micrococcal nuclease (New England Biolabs, Cat. M0247S), vortexed and incubated at 37 °C for 5 min. Chromatin-bound fractions were collected after centrifugation at 18,000×g for 5 min at 4 °C. Total cell lysates and chromatin-bound proteins were immunodetected following SDS-PAGE.

Chromatin-bound fractions were immunoprecipitated using an anti-FLAG antibody (Sigma, F1804) with 4 μg of antibody per sample. After rotation overnight at 4 °C, 70 μl of Sepharose-beads were added and samples were incubated for an additional 1 h at 4 °C with rotation. The protein-bead complexes were collected by centrifugation at 1700×g for 3 min and washed three times with PBS. Seventy μl 2× SDS sample loading dye was added and the complexes were boiled for 10 min. Protein binding was immunodetected following SDS-PAGE. Unprocessed original scans of blots are shown in Supplementary Fig. 6.

The following primary antibodies were used: anti-RepID (NCI186), anti-CUL1 (Abcam, 75817), anti-CUL2 (Abcam, ab166917), anti-CUL3 (Abcam, ab75851), anti-CUL4A (Abcam, ab92554), anti-CUL4B (Sigma, C9995), anti-CUL5 (Abcam, ab184177), anti-CUL7 (Abcam, ab96861), anti-CDT1 (Cell Signaling, 8064S), anti-SET8 (Cell Signaling, 2996), anti-p21 (Santa Cruz, sc-397; Invitrogen, OP64), anti-DDB1 (Abcam, ab109027), anti-DDB2 (Cell Signaling, 5416S), anti-CDT2 (Abcam, ab72264), anti-RBX1 (Cell Signaling, 4397S), anti-SKP2 (Abcam, ab68455), anti-Geminin (Cell Signaling, 5165), anti-PCNA (Santa Cruz, sc-7907), anti-H3K4me3 (Cell Signaling, 9751P), anti-H3K9me3 (Cell Signaling, 13969P), anti-FLAG (Sigma, F1804), anti-α-tubulin (Sigma, T9026), anti-histone H3 (Millipore, 07-690) and anti-γH2AX (Millipore, 05-636). For secondary antibodies, HRP-linked anti-mouse IgG (Cell Signaling, 7076), HRP-linked anti-rabbit IgG (Cell Signaling, 7074) and HRP-linked anti-goat IgG (Santa Cruz, sc-2020) were used following the manufacturer's suggested protocols.

**Data availability.** The data sets for the ChIP-seq and nascent-strand sequencing are available from GEO under Accession code GSE114703. All data within the manuscript is available from the authors upon reasonable request.

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

## Acknowledgements

This study was supported by the Intramural Research Program of the NIH, Center for Cancer Research, National Cancer Institute. We thank Drs Bao Tran, Jyoti Shetty and Yongmei Zhao from the CCR core sequencing facility (FNLCR, Frederick, MD) for help with DNA sequencing. We thank Drs William Gillette, Dominic Esposito and Jane Jones (Protein Expression Laboratory, FNLCR, Frederick, MD) for their help constructing the purified RepID fragments. We are grateful to Dr Langston Lim (Confocal Microscope Core Facility, CCR, NCI, NIH, Bethesda, MD) for providing a productive environment for confocal microscope analyses. We thank Drs. Jean Cook, Anindya Dutta, Zoi Lygerou and Johannes Walter for helpful discussions.

## Author contributions

S.J,. Y.Z., and M.I.A. designed the study. S.J., Y.Z., K.U., H.F., C.E.R., and M.I.A. designed the experiments. S.J., Y.Z., K.U., A.B.M., O.K.S., H.F., C.E.R. and M.I.A. performed the

experiments. S.J., Y.Z., A.B.M., O.K.S., H.F., and C.E.R. analyzed data. S.J., H.F., C.E.R., and M.I.A. wrote the manuscript. S.J., K.U., C.J.R., H.F., C.E.R., A.M.B., D.A.T., and M.I. A. revised the manuscript.

## Additional information

**Competing interests:** The authors declare no competing interests.

