## [Peer Review File · Nature Communications]

Reviewers' comments:

Reviewer #1 (Remarks to the Author):

Jang et al. The Replication-initiation determinant protein (RepID) modulates replication by recruiting CUL4 to chromatin

The authors present novel evidence that RepID regulates CUL4 recruitment to chromatin to control origin firing and prevent re-replication; CUL4 is a member of the CUL4 ubiquitin ligase complex that regulates replication initiation at specific origins by degrading chromatin-bound substrates, and they further demonstrate that the RepID origins are distinct from those regulated by the SCF ubiquitin ligase complex. Both co-IP and western blot analysis of CUL4 recruitment to chromatin demonstrate that the RepID WD40 domain is required for recruitment of CUL4A/B. The authors further demonstrate RepID/CUL4 binding to early S phase origins and CUL4 binding to a distinct set of (later) origins. The authors demonstrate increased re-replication, DNA damage, and reduced viability when RepID depleted cells are treated with the SKP2 inhibitor. These results suggest that disruption of the RepID/CUL4 regulatory system may promote genome instability, and the study would be improved by inclusion metaphase chromosome or micronuclei analysis to address this question. The study suggests an interesting therapeutic strategy, but the discussion is incomplete on this point. In general the conclusions are supported by high quality data, and the presentation is clear, but several issues need attention as noted below.

Fig. 1a: the conclusion that RepID KO reduces chromatin-bound CUL4A/B by at least 80% seems to hold in most cases, but not in HCT116 for CUL4A. This may be due to a technical issue with this blot or a cell-line specific effect, but it needs to be resolved one way or the other.

Fig. 1h: I agree with the conclusion that RepID and CUL4A colocalize mostly in G1 and early S; less colocalization in late S. The authors don't seem to offer a mechanistic explanation for this observation.

Lines 253-255/Fig. 3: I agree that these data indicate that CUL4A is associated with distinct groups of replication origins, but it is unclear how the authors reached the next conclusion, that this differential colocalization is linked to replication timing. While the ChIP-seq pie-charts in fig. 3 distinguish early, mid, late-S origins, and different percentages of early/mid/late origins appear to bound by CUL4A (+/-RepID and CTD2), clear trends are not obvious (nor pointed out by the authors). Is the conclusion that CUL4A+RepID bound origins tend to be early since groups 2 and 4 show the highest early origin values?

Fig. 5c,d: these Venn diagrams show only two overlapping circles, yet four values are reported. Also, the number of peaks and percentages don't seem to correlate: how can 6759 peaks equal 8.6% while 6216 peaks equal 50.9%?

Discussion: the authors indicate (lines 355-356) that defects in CUL4 and/or RepID protein may confer sensitivity to SKP2 inhibition, for example in cancer treatment. In the discussion the authors revisit this idea, but only point out that such defects are associated with other problems (obesity, etc). Is there evidence for CUL4 or RepID defects in cancer? This needs to be discussed to strengthen the case for SKP2i therapy.

Minor issues and typos:

Lines 75 and 96 (and elsewhere): use consistent terminology, ubiquitylation or ubiquitination (both correct, latter is more common).

Line 111-112: ...suggest a role for RepID in establishing and regulating the orderly progression of DNA replication. I think the meaning here is progression of origin firing – as written it is

ambiguous because this phrase can be interpreted as “RepID regulates DNA polymerase progression.”

Fig. 1b: since 3 cell lines are tested, do these bar graphs refer to all three? Or just one line? Fig. 1b suggests that CUL1 chromatin binding is increased 2-fold in RepID KO, but this is only apparent in K562, not in U2OS or HCT116. Does this mean the bar graph refers to K562? It would be helpful to use a more informative label for the y-axis than just ‘fold change’ – it cannot be interpreted without referring to the legend (also true for several other figs, e.g., Sup Fig 2a, 2b, 2c; Sup fig 2d does include an informative axis label). I understand that CUL1 is tangential and the authors do state that the results with non-CUL4 cullins is inconsistent.

Fig. 1h: The excel files presenting the colocalization calculations (Pearson’s coefficients) is useful, but difficult to follow. Explanations of terminology would help (I assume Rcoloc means R values for colocalization). The file would be easier to follow/check if the formulas were included instead of just values. It’s also unclear why only certain measurements were made, i.e., why does WT CUL4A+ FLAG only include G1, early S and late S? why exclude G2/M?

Fig. 3a: typo: non-relicating

Line 279: Consistent with the above results, 70.1% of chromatin-CDT1 was 279 detected in re-replicating plots in RepID KO cells. Is the intended meaning “re-replicating cells”?

Fig. 5b: error bars represent SD, but n is not indicated, and stats are not shown, so it’s impossible to evaluate the significance of the observed differences. Also, the text indicates that chromatin-bound CUL1 increases in RepID KO cells 9-15 h after nocodazole release, but even greater effects are seen 27-33 h after release yet are not mentioned.

Methods:

I did not see information regarding confirmation (or prior publication) of the RepID KO other than western blot. The identity of the mutated alleles (sequence) should be noted.

Line 451: what stain was used to determined DNA content in flow cytometry?

Line 487: confocal microscope

Reviewer #2 (Remarks to the Author):

This manuscript represents a follow up study to a previous publication by this group on the molecular mechanisms by which RepID modulates DNA replication in mammalian cells. The main findings from this manuscript are: 1) RepID recruits the CRL4 complex to chromatin prior to DNA synthesis, indicating a role in the proper licensing of chromosomes for replication. 2) RepID depleted cells relied on an alternative ubiquitin ligase complex (SKP2- containing SCF) to progress through the cell cycle. 3) RepID depletion increased sensitivity to SKP2 inhibitors, which caused genome re-replication. And 4) Both RepID and SKP2 interact with distinct, non-overlapping groups of replication origins. These findings are novel and significant and suggest that interactions of replication origins with specific CRL components execute the DNA replication program and therefore prevent re-initiation of DNA replication.

The experiments are logical, clearly explained and high quality. The conclusions drawn from the results are justified and quite thought provoking. This manuscript represents a significant advance to the field towards the understanding of the replication timing program. I recommend publication without any changes.

Reviewer #3 (Remarks to the Author):

The ubiquitin ligase complexes, CRL4 and SCF, play important roles in the regulation of replication licensing. In this manuscript, the authors demonstrated that the replication origin binding protein RepID, a DCAF that is associated with CRL4, is required for the CRL4 complex to be recruited to chromatin to degrade CRL4 substrates including Cdt1. In RepID KO cells, rereplication is significantly increased. Interestingly, authors showed that more than 70% of CRL4 genomic DNA binding sites are associated with replication origins with significant overlap with the RepID and Cdt2 origin binding sites. RepID-associated CRL4 origin binding sites are enriched at early-replication origins. In RepID KO cells, SKP2-containing SCF binds significantly more to late replication origins to compensate the loss of CRL4 for degradation of Cdt1 to prevent rereplication. Inhibition of RepID markedly increases cellular sensitivity to SKP2 inhibitors, providing a new strategy to use SKP2 inhibitors to treat RepID deficient cancer cells. The findings in this report are interesting and have treatment implication, and thus would be of interest to general audience. However, several points need to be strengthened to support the proposed model.

1. The authors propose that RepID recruits CRL4 to chromatin through RepID and CRL4 interaction. However, the interaction of RepID and Cul4A/Cul4B is very weak (Fig.1D). The endogenous interaction of RepID with Cul4A/Cul4B needs to be shown. Also, it needs to be tested whether the interaction of RepID and CRL is direct, and if so, which components of CRL mediate the direct interaction.

Minor point: The expression of FL-RepID in WT and KO cells is similar (input, right panel), but the amount of RepID in Flag IP is very different (IP-FLAG, left panel, bottom panel)- the IP should be controlled better.

2. The chromatin binding of Cul4B in Fig.1e is too weak and not convincing.

3. In RepID KO cells, the recruitment of CUL4A is greatly reduced (Fig.1g, probably less than 10% left). Cdt2 binds to CRL4, but CUL4A recruitment does not depend on Cdt2 (Suppl. Fig.1a). However, inhibition of CDT2 in RepID KO cells induces significantly more rereplication than that in RepID KO cells (15% vs 29%), suggesting that CDT2 has a major role independent of RepID to suppress rereplication. Since CRL4 chromatin is already minimal in RepID KO cells, it is not clear how Cdt2 functions to suppress rereplication in RepID KO cells.

4. It is interesting that more than 70% of CRL4 genomic DNA binding sites are associated with replication origins. However, it is not clear how this origin binding is related to CRL4 chromatin binding, Cdt1 degradation and rereplication inhibition. Despite a very significant reduction of CRL4 chromatin binding in RepID KO cells, the overlapping binding of Cul4A with RepID to origins is only 15%, while the overlapping binding of Cul4A with Cdt2 to origins is 30%. It raises a question of how origin binding of RepID contributes to CRL4 chromatin binding and degradation of CRL4 substrate such as Cdt1, and what is the underlying mechanism of origin binding to regulate CRL4 chromatin binding. Since Cdt2 origin binding overlaps with CRL4 more than RepID does, it is important to address the role of Cdt2 at origins with CRL4 (which seems not to be required for CRL4 chromatin recruitment).

5. RepID-associated CUL4A DNA binding sites are enriched at early-replication origins, while Cdt2 binding sites are highly colocalized with SKP2 binding sites that mainly replicate late. It would be informative to perform ChIP of CUL4A in RepID KO cells and in Cdt2 deficient cells to see how Cul4A origin binding changes (early vs late origins). The WD40 domain mutants of RepID should be tested to address whether the interaction of RepID and CRL4 is important for CRL4 origin binding.

6. Authors proposed that "Chromatin-recruited CUL4A mediated by RepID may therefore prefer early replication origins, whereas CUL4A recruited in a PCNA-dependent manner may associate with late replication origins." This is supported by differential origin binding of RepID/CUL4A and Cdt2/Cul4A. However, the model needs to be further developed to explain why the majority of CRL4 chromatin recruitment depends on RepID, but not Cdt2.

Reviewer #4 (Remarks to the Author):

In a prior publication, the authors found that the RepID (DACF14, PHIP) protein facilitates origin firing. In this manuscript, the authors show that the RepID protein further regulates replication origins through the recruitment of the CRL4 complex, which prevents re-initiation and spurious DNA replication.

This is an interesting article demonstrating some of the mechanism by which the RepID protein modulates DNA replication and replication timing. The article reads well, and I only have a series of minor suggestions/comments (below), for the authors to consider. I recommend the article be published after minor revisions.

Minor comments:

- Figure 1D: In page 8, the statement is made that 'results clearly show that CUL4A/B recruitment to chromatin is mediated by an interaction with the WD40 domain of RepID'.

The N-terminal truncations indeed lead to a loss of association between the two proteins and do suggest that the N-terminal half of the protein is involved in the interaction. However, we still do not know whether the interaction between RepID and CUL4 is a direct one between RepID WD40 repeats and CUL4 or if it's indirectly mediated by other proteins (e.g. does recombinant RepID WD40 repeats bind recombinant CUL4).

- The authors may want to indicate the number of cells that were analyzed in the figure legends (when appropriate such as in Figure 1H). Error bars obtained from experimental triplicates should also be included as much as possible (e.g. Figures 2F, 3A).

- Figure 1B: Is the quantification in this figure representing results from all three different cell lines? The mixing of data from different cell lines may lead to confusion and quantifications (if kept in the manuscript) should show experimental triplicates for each of the three separate cell lines (no mixing of experimental systems/cell lines).

- Suppl. Figure 1C: The CDT2 blot is difficult to interpret and would be best left out.

- Figure 2C: Why is there so much CUL4A signal in the top-left RepID KO, siCUL4A+B immunofluorescence? Is this representative of what you saw? If so, how specific is this CUL4A antibody by IF (the knockdowns seem >80% effective by western)?

- Suppl. Figure 2D is not convincing (since the CDT1 bands for wt and RepID KO seem to be on separate blots, so the exposure levels may not be the same). The experiment should either be repeated (including p21 and SET8, to be consistent), or removed.

- Would the RepID + CUL4A ChIP-Seq peaks correlation with histone marks be higher if looking at histone acetylation (instead of H3K4me3), since the protein does have a bromodomain? How does it also compare to H4K20me1 (the SET8 product that is also linked to origins of replication)?

- Line 388: rephrase to clarify that EED is a subunit of the PRC2 methyltransferase (not the actual enzyme, and certainly not catalyzing H3K9 methylation).

Point by point response to reviewers comments:

We thank all the reviewers for the thoughtful evaluation of our original submission and for the helpful suggestions. We have revised the manuscript based on the reviewers' comments, and we believe that the revision has improved the paper and clarified some important points. We greatly appreciate the reviewers' time and help. Below is a detailed description of how we have addressed each of the comments:

Reviewer #1 (Remarks to the Author):

Jang et al. The Replication-initiation determinant protein (RepID) modulates replication by recruiting CUL4 to chromatin

Comment: *The authors present novel evidence that RepID regulates CUL4 recruitment to chromatin to control origin firing and prevent re-replication; CUL4 is a member of the CRL4 ubiquitin ligase complex that regulates replication initiation at specific origins by degrading chromatin-bound substrates, and they further demonstrate that the RepID origins are distinct from those regulated by the SCF ubiquitin ligase complex. Both co-IP and western blot analysis of CUL4 recruitment to chromatin demonstrate that the RepID WD40 domain is required for recruitment of CUL4A/B. The authors further demonstrate RepID/CUL4 binding to early S phase origins and CUL1 binding to a distinct set of (later) origins. The authors demonstrate increased re-replication, DNA damage, and reduced viability when RepID depleted cells are treated with the SKP2 inhibitor. These results suggest that disruption of the RepID/CUL4 regulatory system may promote genome instability, and the study would be improved by inclusion metaphase chromosome or micronuclei analysis to address this question.*

Answer:

We thank the reviewer for raising this point and for the suggestion. The revised submission contains micronuclei analyses using three RepID WT and KO cell lines. The results are indeed consistent with the interpretation that loss of RepID/CUL4A promotes genome instability. Genomic instability in cells lacking RepID is also supported by our observation that γ H2AX and 53BP1 foci formation increases in RepID KO cells (Supplementary fig. 2e-g).

Comment: *The study suggests an interesting therapeutic strategy, but the discussion is incomplete on this point. In general the conclusions are supported by high quality data, and the presentation is clear, but several issues need attention as noted below.*

Answer:

We thank the reviewer for this evaluation. We have followed the reviewer's advice below to include a discussion of the issues mentioned by the reviewer.

Comment: Fig. 1a: the conclusion that RepID KO reduces chromatin-bound CUL4A/B by at least 80% seems to hold in most cases, but not in HCT116 for Cul4A. This may be due to a technical issue with this blot or a cell-line specific effect, but it needs to be resolved one way or the other.

Answer:

We thank the reviewer for this comment. In the revision, we replaced the original submitted figure, which showed a high background, with a lower exposure of CUL4A analysis in HCT116 cells (Fig. 1a). The new data show that in HCT116 cells, RepID KO associates with over 80% reduction of chromatin-bound CUL4A/B.

Comment: Fig. 1h: I agree with the conclusion that RepID and CUL4A colocalize mostly in G1 and early S; less colocalization in late S. The authors don't seem to offer a mechanistic explanation for this observation.

Answer:

We thank the reviewer for this comment. The decreased colocalization between RepID and CUL4A in late S phase (Fig. 1h) correlates with the observed decreased chromatin binding of RepID (Fig. 1g), suggesting that RepID and CUL4A only interact when both proteins are on chromatin. The revised manuscript mentions the concordance between colocalization and CUL4A chromatin levels in the context of the results of Figure 1h (page 9 line 6) and extensively discussed the timing of CUL4A chromatin association in the Discussion section. (Also please see our responses to the other reviewers, below).

Comment: Lines 253-255/Fig. 3: I agree that these data indicate that CUL4A is associated with distinct groups of replication origins, but it is unclear how the authors reached the next conclusion, that this differential colocalization is linked to replication timing. While the ChIP-seq pie-charts in fig. 3 distinguish early, mid, late-S origins, and different percentages of early/mid/late origins appear to bound by CUL4A (+/-RepID and CTD2), clear trends are not obvious (nor pointed out by the authors). Is the conclusion that CUL4A+RepID bound origins tend to be early since groups 2 and 4 show the highest early origin values?

Answer:

We thank the reviewer for this comment. Yes, the reviewer is correct that the interactions of groups 2 and 4 (Fig. 3) with early replicating chromatin gave us the first clue regarding differential enrichment of RepID and CDT2 in discrete replication timing domains. The reviewer's comment prompted us to clarify the relationship further, and in the revision we have depicted the distribution of RepID and CDT2 bound origins in early- and late-replicating chromatin (Supplementary Fig. 3b) and provided an expanded analysis of chromatin associations with all groups of CUL4, CDT2 and RepID bound origins (Fig. 3c,d and Supplementary Fig. 3). These analyses show that RepID bound origins are enriched in early replicating chromatin whereas CDT2 bound origins replicate throughout the S-phase, and that RepID bound origins highly colocalize with histone marks associated with early replication (H3K4me3, H3K9Ac, H3H27Ac), whereas CDT2 bound origins colocalize with the late replicating chromatin mark H3K9me3.

Comment: Fig. 5c,d: these Venn diagrams show only two overlapping circles, yet four values are reported. Also, the number of peaks and percentages don't seem to correlate: how can 6759 peaks equal 8.6% while 6216 peaks equal 50.9%?

Answer:

We apologize for the confusing representation. The four values in the original Fig. 5e represented the percentage of colocalization as a fraction of replicating origins as well as a fraction of SKP2 peaks. We agree that the multiple values are confusing, and we have replaced the Venn diagrams by a table showing all the data (Fig. 5c) and the original colocalization plots (Fig. 5e,f).

Comment: Discussion: the authors indicate (lines 355-356) that defects in CUL4 and/or RepID protein may confer sensitivity to SKP2 inhibition, for example in cancer treatment. In the discussion the authors revisit this idea, but only point out that such defects are associated with other problems (obesity, etc). Is there evidence for CUL4 or RepID defects in cancer? This needs to be discussed to strengthen the case for SKP2i therapy.

Answer: Thank you for pointing out this important omission. The revised version explicitly states that both CUL4 and RepID modulate cancer survival, NEDDylation inhibitors targeting CRLs are currently in clinical trials, and that SKP2 inhibitors are in development. In addition, we state upfront in the Introduction section (page 5, second paragraph line 6) that RepID is a known marker of melanoma aggressiveness and CUL4 is overexpressed in a series of cancers. We discuss this point again in the penultimate and last paragraphs of the Discussion section, pages 20-21.

Comment: Minor issues and typos:

Lines 75 and 96 (and elsewhere): use consistent terminology, ubiquitylation or ubiquitination (both correct, latter is more common).

Answer:

We thank the reviewer for this comment. We changed “ubiquitylation” to “ubiquitination” throughout the text.

Comment: Line 111-112: ...suggest a role for RepID in establishing and regulating the orderly progression of DNA replication. I think the meaning here is progression of origin firing – as written it is ambiguous because this phrase can be interpreted as “RepID regulates DNA polymerase progression.”

Answer:

Thank you, we agree. We modified the text from “progression of DNA replication” to “progression of replication origin activation” (page 6 line 2 in the revised text).

Comment: Fig. 1b: since 3 cell lines are tested, do these bar graphs refer to all three? Or just one line? Fig. 1b suggests that CUL1 chromatin binding is increased 2-fold in RepID KO, but this is only apparent in K562, not in U2OS or HCT116. Does this mean the bar graph refers to K562? It would be helpful to use a more informative label for the y-axis than just ‘fold change’ – it cannot be interpreted without referring to the legend (also true for several other figs, e.g., Sup Fig 2a, 2b, 2c; Sup fig 2d does include an informative axis label). I understand that CUL1 is tangential and the authors do state that the results with non-CUL4 cullins is inconsistent.

Answer:

Thank you, we agree, to clarify this point we included data from all 3 individual cell lines in the revised Fig. 1b. We have also changed the labeling on the y-axis from “Fold Changes” to “Relative Density”.

Comment: Fig. 1h: The excel files presenting the colocalization calculations (Pearson’s coefficients) is useful, but difficult to follow. Explanations of terminology would help (I assume Rcoloc means R values for colocalization). The file would be easier to follow/check if the formulas were included instead of just values. It’s also unclear why only certain measurements were made, i.e., why does WT CUL4A+ FLAG only include G1, early S and late S? why exclude G2/M?

Answer:

Thank you, the file was modified and the paper revised paper includes information on Pearson’s coefficients in the Methods section, Page 24 line 13. We excluded G2/M because at that time, cells do not incorporate EdU and data on the colocalization between CUL4 and EdU would not be informative.

Comment: Fig. 3a: typo: non-replicating

Answer:

Thanks, this was corrected.

Comment: Line 279: Consistent with the above results, 70.1% of chromatin-CDT1 was 279 detected in re-replicating plots in RepID KO cells. Is the intended meaning “re-replicating cells”?

Answer:

The reviewer is correct. We have revised accordingly.

Comment: Fig. 5b: error bars represent SD, but n is not indicated, and stats are not shown, so it's impossible to evaluate the significance of the observed differences.

Answer:

We thank the reviewer for pointing this out. In the revision, we have labeled the graphs with indicators of statistical significance as shown in the figure legend.

Comment: Also, the text indicates that chromatin-bound CUL1 increases in RepID KO cells 9-15 h after nocodazole release, but even greater effects are seen 27-33 h after release yet are not mentioned.

Answer:

Thank you for this observation, we agree. We think that the increase in chromatin-bound CUL1 in RepID KO after the second S-phase of proliferation without RepID represents an adaption to utilize the SCF instead of RepID (in the revision, this issue is discussed on page 15 line 3-5).

Comment: Methods:

I did not see information regarding confirmation (or prior publication) of the RepID KO other than western blot. The identity of the mutated alleles (sequence) should be noted.

Answer: We apologize for this omission. The information is included in the revision (Methods page 22 second paragraph and Supplementary Fig. 1a).

Comment: Line 451: what stain was used to determined DNA content in flow cytometry?

Answer:

Sorry for the omission, we used DAPI as a DNA stain. We added this information to the revised manuscript (Methods section page 23, third paragraph line 3).

Comment: Line 487: confocal microscope

Answer:

Corrected, thanks.

Reviewer #2 (Remarks to the Author):

Comment: This manuscript represents a follow up study to a previous publication by this group on the molecular mechanisms by which RepID modulates DNA replication in mammalian cells. The main findings from this manuscript are: 1) RepID recruits the CRL4 complex to chromatin prior to DNA synthesis, indicating a role in the proper licensing of chromosomes for replication. 2) RepID depleted cells relied on an alternative ubiquitin ligase complex (SKP2- containing SCF) to progress through the cell cycle. 3) RepID depletion increased sensitivity to SKP2 inhibitors, which caused genome re-replication. And 4) Both RepID and SKP2 interact with distinct, non-overlapping groups of replication origins. These findings are novel and significant and suggest that interactions of replication origins with specific CRL components execute the DNA replication program and therefore prevent re-initiation of DNA replication.

The experiments are logical, clearly explained and high quality. The conclusions drawn from the results are justified and quite thought provoking. This manuscript represents a significant advance to the field towards the understanding of the replication timing program. I recommend publication without any changes.

Answer:

We thank the reviewer for the positive evaluation of the manuscript and the supportive comments.

Reviewer #3 (Remarks to the Author):

Comment: The ubiquitin ligase complexes, CRL4 and SCF, play important roles in the regulation of

replication licensing. In this manuscript, the authors demonstrated that the replication origin binding protein RepID, a DCAF that is associated with CRL4, is required for the CRL4 complex to be recruited to chromatin to degrade CRL4 substrates including Cdt1. In RepID KO cells, rereplication is significantly increased. Interestingly, authors showed that more than 70% of CRL4 genomic DNA binding sites are associated with replication origins with significant overlap with the RepID and Cdt2 origin binding sites. RepID-associated CRL4 origin binding sites are enriched at early-replication origins. In RepID KO cells, SKP2-containing SCF binds significantly more to late replication origins to compensate the loss of CRL4 for degradation of Cdt1 to prevent rereplication.

Answer:

We thank the reviewer for this evaluation and the suggestions below.

Comment: 1. The authors propose that RepID recruits CRL4 to chromatin through RepID and CRL4 interaction. However, the interaction of RepID and Cul4A/Cul4B is very weak (Fig.1D). The endogenous interaction of RepID with Cul4A/Cul4B needs to be shown. Also, it needs to be tested whether the interaction of RepID and CRL is direct, and if so, which components of CRL mediate the direct interaction.

Minor point: The expression of FL-RepID in WT and KO cells is similar (input, right panel), but the amount of RepID in Flag IP is very different (IP-FLAG, left panel, bottom panel)- the IP should be controlled better.

Answer:

We thank the reviewer for pointing this out. We have repeated the experiments and the revised figure is shown in the revision, Fig. 1d. In addition, we have performed a pull-down of endogenous CUL4A/4B and in the revision one can see clearly that RepID can bind CUL4A/4B (Supplementary Fig. 1c). In vitro interaction studies also clearly showed that RepID can interact with CRL4 and DDB1 directly using its WD40 domain. We have included this result at the revised submission (Supplementary Fig. 1d).

Comment: 2. The chromatin binding of Cul4B in Fig.1e is too weak and not convincing.

Answer:

We apologize for the unclear image. In the revision, we have performed the experiment again in triplicate and included a new image (Fig. 1e).

Comment: 3. In RepID KO cells, the recruitment of CUL4A is greatly reduced (Fig.1g, probably less than 10% left). Cdt2 binds to CRL4, but CUL4A recruitment does not depend on Cdt2 (Suppl. Fig.1a). However, inhibition of CDT2 in RepID KO cells induces significantly more rereplication than that in RepID KO cells (15% vs 29%), suggesting that CDT2 has a major role independent of RepID to suppress rereplication. Since CRL4 chromatin is already minimal in RepID KO cells, it is not clear how Cdt2 functions to suppress rereplication in RepID KO cells.

Answer:

We thank the reviewer for this comment, we agree that the dynamic interaction between the two DCAFs, CDT2 and RepID, is an important point. We hypothesize that the increased sensitivity to CDT2 depletion reflects an adaptation in RepID KO cells to progress through the cell cycle with low levels of chromatin-bound CUL4A, increasing the dependence on CDT2. This issue is mentioned in the revision (page 11, second paragraph lines 11-12;also please see the answer to point #4, below).

Comment: 4. It is interesting that more than 70% of CRL4 genomic DNA binding sites are associated with replication origins. However, it is not clear how this origin binding is related to CRL4 chromatin binding, Cdt1 degradation and rereplication inhibition. Despite a very significant reduction of CRL4 chromatin binding in RepID KO cells, the overlapping binding of Cul4A with RepID to origins is only 15%, while the overlapping binding of Cul4A with Cdt2 to origins is 30%. It raises a question of how origin binding of RepID contributes to CRL4 chromatin binding and degradation of CRL4 substrate such as Cdt1, and what is the underlying mechanism of origin binding to regulate CRL4 chromatin binding. Since Cdt2 origin binding overlaps with CRL4 more than RepID does, it is important to address the role of Cdt2 at origins with CRL4 (which seems not to be required for CRL4 chromatin recruitment).

Answer:

We thank the reviewer for raising this point, and we discuss it in detail in the revision (the first and the penultimate paragraphs of the Discussion section, pages 18 and 20-21). Our data suggest that although RepID binding is only evident at a subgroup of replication origins, CUL4A recruitment to all locations on chromatin during G1 requires RepID. It is possible that RepID recruits CUL4A to chromatin and then remains bound to a fraction of the potential replication origins. In studies that are beyond the scope of the current paper, we are investigating the possibility that the higher overlap between CDT2 and CUL4A might reflect a higher stability or a longer chromatin retention of the CDT2-CUL4 interaction, possibly through the involvement of PCNA.

Comment: 5. RepID-associated CUL4A DNA binding sites are enriched at early-replication origins, while Cdt2 binding sites are highly colocalized with SKP2 binding sites that mainly replicate late. It would be informative to perform ChIP of CUL4A in RepID KO cells and in Cdt2 deficient cells to see

how Cul4A origin binding changes (early vs late origins). The WD40 domain mutants of RepID should be tested to address whether the interaction of RepID and CRL4 is important for CRL4 origin binding.

Answer:

We thank the reviewer for these suggestions. Indeed, the WD40 domain of RepID was required for CUL4 recruitment to chromatin (Fig. 1e; both CUL4A and CUL4B associate with chromatin only in cells that contain forms of RepID that include the WD40 domain). We had performed ChIP with CUL4A in RepID KO cells as suggested, however the results confirmed our observation that the level of CUL4A binding in RepID KO was very low and we were not able to obtain clear peaks for further analyses. Similarly, CUL4A ChIP in CDT2 deficient cells could not be performed due to the severe DNA damage and cell death in cells with transient CDT2 deficiency (SubG1 in Fig. 3a). Because CUL4 is known to be recruited to DNA damage sites (for example, see Ishii et al., JBC 285, 41993-42000), we expect that the results of a CUL4A ChIP-seq in cells with CDT2 deficiencies might reflect DNA damage hotspots, not replication origins.

Comment: 6. Authors proposed that “Chromatin-recruited CUL4A mediated by RepID may therefore prefer early replication origins, whereas CUL4A recruited in a PCNA-dependent manner may associate with late replication origins.” This is supported by differential origin binding of RepID/CUL4A and Cdt2/Cul4A. However, the model needs to be further developed to explain why the majority of CRL4 chromatin recruitment depends on RepID, but not Cdt2.

Answer:

We thank to referee’s comment and for the suggestion to develop the model further. In the revision, we include an expanded discussion of the model (Fig. 6 – the graphical depiction of the model did not change to avoid introducing complexity). We propose (page 18 lines 12-16) that CRL4 chromatin recruitment by RepID occurs during the G1 phase of the cell cycle, concomitant with assembly, whereas the interactions with CDT2 occur later and are evident in origins that had not recruited RepID and CRL4. Although recruitment of CUL4 to chromatin during G1 depends on RepID, the degradation of CDT1 during S-phase requires both CDT2 and RepID. Because we found that SKP2 associates with chromatin during late S-phase, we also propose that SKP2-containing SCF can take over some of the functions of CRL4 in residual origins that contain CDT1 after dissociation of RepID.

Reviewer #4 (Remarks to the Author):

Comment: In a prior publication, the authors found that the RepID (DCAF14, PHIP) protein facilitates origin firing. In this manuscript, the authors show that the RepID protein further regulates replication origins through the recruitment of the CRL4 complex, which prevents re-initiation and spurious DNA replication.

This is an interesting article demonstrating some of the mechanism by which the RepID protein modulates DNA replication and replication timing. The article reads well, and I only have a series of minor suggestions/comments (below), for the authors to consider. I recommend the article be published after minor revisions.

Answer:

We thank the reviewer for this evaluation and the comments below.

Comment: Minor comments:

- Figure 1D: In page 8, the statement is made that 'results clearly show that CUL4A/B recruitment to chromatin is mediated by an interaction with the WD40 domain of RepID'.

The N-terminal truncations indeed lead to a loss of association between the two proteins and do suggest that the N-terminal half of the protein is involved in the interaction. However, we still do not know whether the interaction between RepID and CUL4 is a direct one between RepID WD40 repeats and CUL4 or if it's indirectly mediated by other proteins (e.g. does recombinant RepID WD40 repeats bind recombinant CUL4).

Answer: We thank the reviewer for pointing this out. In the revision, we include an in vitro binding assay using purified proteins demonstrating a direct interaction between RepID WD40 domain and DDB1 and CUL4 (Supplementary Fig. 1d).

Comment: - The authors may want to indicate the number of cells that were analyzed in the figure legends (when appropriate such as in Figure 1H). Error bars obtained from experimental triplicates should also be included as much as possible (e.g. Figures 2F, 3A).

Answer:

We thank the reviewer for pointing out this omission. In the revision, we added the quantification as suggested (Fig. 1h). We have also added a clarification that the results shown in Fig. 2f, 3a were average from three independent biological experiments and added error bars to the revised Fig. 2f, 3a.

Comment: - Figure 1B: Is the quantification in this figure representing results from all three different cell lines? The mixing of data from different cell lines may lead to confusion and quantifications (if kept in the manuscript) should show experimental triplicates for each of the three separate cell lines (no mixing of experimental systems/cell lines).

Answer:

Thank you, again we agree and apologize for the confusion. We have revised Fig. 1b to show data from each individual cell line, as was also suggested by Reviewer #1's comment.

Comment: - Suppl. Figure 1C: The CDT2 blot is difficult to interpret and would be best left out.

Answer:

We agree and thank the reviewer for pointing this out. In the revision, we left out the CDT2 blot.

Comment: - Figure 2C: Why is there so much CUL4A signal in the top-left RepID KO, siCUL4A+B immunofluorescence? Is this representative of what you saw? If so, how specific is this CUL4A antibody by IF (the knockdowns seem >80% effective by western)?

Answer:

Thank you for pointing this out. In the revision, the slides were re-imaged using an identical threshold (Minimum : 200, Maximum : 65535) to prevent saturation (Fig. 2c and Supplementary Fig. 1g). The specificity of the CUL4A antibody was further tested using immunoblotting, and the CUL4A protein signal markedly decreased in siRNA-CUL4A treated cells (revised Fig. 2c).

Comment: - Suppl. Figure 2D is not convincing (since the CDT1 bands for wt and RepID KO seem to be on separate blots, so the exposure levels may not be the same). The experiment should either be repeated (including p21 and SET8, to be consistent), or removed.

Answer:

We thank the reviewer for pointing this out. As suggested, we repeated experiments and the results are shown in the revised Fig. 2e and Supplementary Fig. 2a,b.

Comment: - Would the RepID + CUL4A ChIP-Seq peaks correlation with histone marks be higher if looking at histone acetylation (instead of H3K4me3), since the protein does have a bromodomain? How does it also compare to H4K20me1 (the SET8 product that is also linked to origins of replication)?

Answer:

We thank the reviewer for this suggestion. In the revision, we include the results of a ChIP-seq experiment with acetylated histone H3K9, H3K27 in K562 cells (Supplementary Fig. 3a). We have not used monomethylated H4K20 in colocalization analyses, however, because H4K20Me1

is very abundant, covering more than 80% of the genome, and peaks generated from ChIP-Seq with this modification were not informative in colocalization analyses. CUL4A peaks colocalizing with RepID (Group 2, 4) shows much higher colocalization with acetylated histones than the entire population of CUL4A peaks or CUL4A peaks colocalizing with CDT2 (Group 1, 3). These observations are consistent with the interpretation that RepID will recognize regions of acetylated histone (also known as early replicating origins), possibly using its bromodomain. In the revision, we also included comparisons between RepID- or CDT2-specific peaks and modified histone H3 or origins stratified by replication timing (Fig. 3b-d and Supplementary Fig. 3)

Comment: - Line 388: rephrase to clarify that EED is a subunit of the PRC2 methyltransferase (not the actual enzyme, and certainly not catalyzing H3K9 methylation).

Answer:

We thank the reviewer and we corrected this error in the revised manuscript.

In summary, we would like to thank all the reviewers again for their thorough evaluation and constructive comments. We feel that the comments helped to improve the manuscript considerably, and we are grateful for the reviewers' time and help.

REVIEWERS' COMMENTS:

Reviewer #1 (Remarks to the Author):

In this revision the authors have significantly improved the report by text revision and by adding new, high-quality data that demonstrate that RepID regulates CUL4 recruitment to chromatin to control origin firing and prevent re-replication.

The authors have done an excellent job responding to all of my prior comments and concerns. My reading also indicates that the authors have taken the same level of care in addressing the concerns of the other reviewers. In summary, this report describes a solid, data-rich study that supports a detailed model of replication origin regulation and the severe consequences when these regulatory systems are defective, with important implications for cancer diagnostics and treatment strategies.

There were a few minor typos and suggested edits:

Line 403: ...suggest that RepID mediates the recruitment...

Line 421: ... H3K27Ac-containing

Line 462: ... proposing a novel role for RepID in recruiting CUL4 to chromatin... Should this be written as "suggesting a novel role"?

Reviewer #3 (Remarks to the Author):

The authors have adequately addressed the questions that I raised. The manuscript is improved.

Reviewer #4 (Remarks to the Author):

Again, this is a highly interesting article that will be of interest to readers of Nature Communications.

There are a few typos left (e.g. line 276 'exhibt'), and there is no reference to Suppl. Fig. 4 in the manuscript (easy corrections to make).

Missing sample numbers and the number of biological replicates have been clarified for the few specific examples listed by the reviewers, but there are still a few experiments in the manuscript for which the sample number and/or number of biological replicates is not clear (again, easy modifications to make prior to the manuscript's publication).

Point by point response to referees' comments:

We thank all the referees for their thoughtful re-evaluation of our manuscript. We were gratified to learn that the revision had addressed most of the concerns. Below is a description of how we have addressed the remaining comments:

Reviewer #1 (Remarks to the Author):

In this revision the authors have significantly improved the report by text revision and by adding new, high-quality data that demonstrate that RepID regulates CUL4 recruitment to chromatin to control origin firing and prevent re-replication.

The authors have done an excellent job responding to all of my prior comments and concerns. My reading also indicates that the authors have taken the same level of care in addressing the concerns of the other reviewers. In summary, this report describes a solid, data-rich study that supports a detailed model of replication origin regulation and the severe consequences when these regulatory systems are defective, with important implications for cancer diagnostics and treatment strategies.

There were a few minor typos and suggested edits:

Line 403: ...suggest that RepID mediates the recruitment...

Line 421: ... H3K27Ac-containing

Line 462: ... proposing a novel role for RepID in recruiting CUL4 to chromatin... Should this be written as "suggesting a novel role"?

Response: *We thank the reviewer for the evaluation and for the suggestions. All the typos have been corrected as detailed in the table below.*

Reviewer #3 (Remarks to the Author):

The authors have adequately addressed the questions that I raised. The manuscript is improved.

Response: *We thank the reviewer for the evaluation and for the suggestions.*

Reviewer #4 (Remarks to the Author):

Again, this is a highly interesting article that will be of interest to readers of Nature Communications.

There are a few typos left (e.g. line 276 'exhibt'), and there is no reference to Suppl. Fig. 4 in the manuscript (easy corrections to make).

Missing sample numbers and the number of biological replicates have been clarified for the few specific examples listed by the reviewers, but there are still a few experiments in the manuscript for which the sample number and/or number of biological replicates is not clear (again, easy modifications to make prior to the manuscript's publication).

Response: *We thank the reviewer for the evaluation and for the suggestions. All the typos have been corrected as detailed in the table below. We have also indicated the number of replicates where the information was missing, as suggested.*

Table of corrections:

previous		revised	
abstract	155 words		150 words
Page 18 (#1-1)	mediate		mediates
Page 19 (#1-2)	containg		containing
Page 21 (#1-3)	proposing		suggesting
Page 13 (#4-1)	exhibt		exhibit
(#4-2)	No ref (Supplementary Fig. 4)		Added: last line, page 13
(#4-3)	Sample number of biological replicates should be indicated		Added
Order	Title - abstract - introduction - results - discussion - methods - figure legends - endnotes - references		Title - abstract - introduction - results - discussion - methods - references - endnotes - figure legends
Page 24	Supplementary file 1		Supplementary Data 1
Results	Subheadings (exceed 60 characters)		Subheading (less than 60 characters)
Methods	rpm		g
Methods	Subheadings (exceed 60 characters)		Subheadings(less than 60 characters)
Methods	Missing Data availability		Added Data availability
Figure legends (length)	Figure 1 : 375 words		348 words
	Figure 2 : 322 words		328 words
	Figure 3 : 187 words		194 words
	Figure 4 : 220 words		224 words
	Figure 5 : 390 words		350 words

In sum, we would like to thank the reviewers again. The reviewers' comments and suggestions have improved the paper and we appreciate their time and help.